



**1**  **High-resolution Beijing MST radar detection of tropopause structure and**

**2**  **variability over Xianghe (39.75° N, 116.96° E), China**

Feilong Chen[1], Gang Chen[1*], Yufang Tian[2], Shaodong Zhang[1], Kaiming Huang[1],
Chen Wu[1], Weifan Zhang[1]
[1]School of Electronic Information, Wuhan University, Wuhan 430072, China.
[2]Key Laboratory of Middle Atmosphere and Global Environment Observation, Institute
of Atmospheric Physics, Chinese Academy of Sciences, Beijing 100029, China.
*Corresponding author: Gang Chen (g.chen@whu.edu.cn)
**Abstract.**
As a result of partial specular reflection from the atmospheric stable layer, the radar
tropopause (RT) can simply and directly be detected by VHF radars with vertical
incidence. Here, the Beijing MST radar measurements are used to investigate the
structure and the variabilities of the tropopause in Xianghe, China with a temporal
resolution of 0.5 hour from November 2011 to May 2017. High-resolution radar-
derived tropopause is compared with the thermal lapse-rate tropopause (LRT) that
defined by the World Meteorological Organization (WMO) criterion from twice daily
radiosonde soundings and with the dynamical potential vorticity tropopause (PVT) that
defined as the height of 2 PVU surface. During all the seasons, the RT and the LRT in
altitude agree well with each other with a correlation coefficient of $\geq 0.74$. Statistically,
weaker (higher) tropopause sharpness seems to contribute to larger (smaller) difference
between the RT and the LRT in altitude. The RT agrees well with the PVT in altitude
during winter and spring with a correlation coefficient of $\geq 0.72$, while the correlation





coefficient in summer is only 0.33. As expected, the monthly mean RT and LRT height
both show seasonal variations. Lomb-Scargle periodograms show that the tropopause
exhibits obvious diurnal variation throughout the seasons, whereas the semidiurnal
oscillations are rare and occasionally observed during summer and later spring. Our
study shows the good capability of the Beijing MST radar to determine the tropopause
height, as well as present its diurnal oscillations.
**Key words:** VHF radar; MST radar; tropopause; diurnal oscillation.
**1.  Introduction**

The tropopause marks a transition zone separating the well-mixed convectively

active troposphere from the stably stratified and more quiescent stratosphere. Its
structure and variability is characterized by large changes in thermal (e.g., lapse rate),
dynamical (e.g., potential vorticity), and chemical properties (e.g., ozone and water
vapor) and hence acts as a key role for the stratosphere-troposphere exchange (STE)
processes (Hoinka, 1998; Seidel et l., 2001). The height of the tropopause depends
significantly on the latitude, with about 17 km near the equator and less than 9-10 km
at polar latitudes (Ramakrishnan, 1933). Over subtropical latitudes with the presence
of subtropical jet, where the tropopause experiences rapid change or breaking,
tropopause folding events are commonly observed (Pan et al., 2004). Climatologically,
the altitude of the tropopause represents the seasonal variation of the flux of
stratospheric air intruding into the troposphere (Appenzeller et al., 1996). Moreover,
the tropopause height trends can be a sensitive indicator of anthropogenic climate





change (Sausen and Santer, 2003; Santer et al., 2003a; Añel et al., 2006).
A variety of ways are available to determine the extratropical tropopause.
Radiosonde sounding is the most commonly used to define the thermal tropopause
(hereafter referred to as LRT) based on temperature lapse-rate (WMO, 1957). The
thermal definition of tropopause can be applied globally and the tropopause height
easily be determined from one individual profile (Santer et al., 2003). Radiosonde
sounding, however, is impracticable in severe weather conditions such as intense
rainfall and cold air outbreak. Another feasible definition is to use a specific potential
vorticity (PV) surface to represent the dynamical tropopause (hereafter referred to as
PVT) (Reed, 1955; Hoskins et al., 1985). Dynamical definition has the advantage that
the PV is a conserved property (under adiabatic and friction-less conditions) of an air
mass (Hoskins et al., 1985; Bethan et al., 1996). Values in the range 1-4 PVU (1 PVU=
$10^6\ m^2\ s^{-1}\ K\ kg^{-1}$) are used in previous researches in the Northern Hemisphere
(e.g. Baray et al., 2000; Sprenger et al., 2003; Hoerling et al., 1991). The threshold of
2 PVU surface is the most commonly used (Gettelman et al., 2011). Dynamical
definition, however, is not applicable near the equator, where the PV tends to be 0 (e.g.,
Hoerling et al., 1991; Nielsen-Gammon et al., 2001).
As a result of partial specular reflection from stable atmospheric layer, the radar
tropopause (RT) can be well represented and identified by atmospheric radars operating
at meter wavelength (VHF band) and directing at vertical incidence (Gage and Green,
1979). Research activity increased remarkably following the first report on VHF radar
detection of tropopause by Gage and Green (1979), for instance, the researches in



middle latitudes (e.g. Hermawan et al., 1998), polar regions (e.g. Hall, 2013a), and
tropical regions (e.g. Das et al., 2008; Ravindrababu et al., 2014). Several methods have
been proposed to determine the tropopause height via radar echo power, including the
largest gradient in echo power (Vaughan et al., 1995; Alexander et al., 2012), the
maximum echo power (Vaughan et al., 1995; Hall et al., 2009), and the specific value
of echo power (Gage and Green, 1982; Yamamoto et al., 2003). The method of the RT
height determination used in this paper will be described in detail in next section.

The biggest advantage of the VHF radar measurements is the ability of continuous

operation unmanned in any weather conditions. Of course, no definition of the
tropopause is perfect. VHF radar system can only be limited to a few locations globally.
A detailed review of the close relationship between these different tropopause
definitions is provided by Alexander et al., (2012).

By means of the radiosonde, reanalysis, and satellite data available globally, long-

term (annual or longer) variability in tropopause height has received extensive attention
(e.g. Randel et al., 2000; Angell and Korshover, 2009; Son et al., 2011; Liu et al., 2014).
However, short period (diurnal or semidiurnal) variability of the tropopause is hard to
be examined by these measurements. In contrast, benefiting from the much higher
temporal resolution, radar definition of the tropopause provides good capability for
studying the diurnal and semidiurnal variation in tropopause height. Earlier, Yamamoto
et al., (2003) reported the capability of the Equatorial Atmospheric Radar to examine
the diurnal variation of tropopause height. Then, the diurnal variability of the tropical
tropopause was investigated in detail by Das et al., (2008) using the Indian Gadanki



MST radar. Its diurnal variation over a polar latitude station was investigated by Hall
(2013b). In the absence of pressure and temperature parameters, the evidence of
atmospheric tides can be well represented by winds (e.g. Huang et al., 2015).

The tropopause structure in midlatitudes is different from that in other regions.

Double tropopauses structure is a ubiquitous feature over mid-latitude regions near
40°N (Pan et al., 2004; Randel et al., 2007). Strong evidence has revealed that the
poleward intrusion of subtropical tropospheric air that occurred above the subtropical
jet have resulted in the double structure (Pan et al., 2009). The higher part (second
tropopause near ~16 km) is characterized by tropical features of cold and higher level,
whereas the lower part (first tropopause near ~12 km) is characterized by polar features
of warm and lower level. In the present study, we focus on the first tropopause which
will be referred to as 'tropopause' hereafter.
So far, knowledge on the high temporal resolution (within 1 hour) structure and
variability of the midlatitude tropopause is still insufficient. In this study, using more
than 5 years of Beijing MST radar echo power measurements in vertical beam, we
mainly focus on the high-resolution characteristics of the tropopause structure and their
comparison with the simultaneous radiosonde and dynamical definitions. Another
important objective of this study is to examine the diurnal and semidiurnal variability
of the tropopause. The observational characteristics of e.g. winds, echo power, and data
acquisition rate near the tropopause layer are also presented in the paper.

**2.  Data and Methods**



### 2.1. Radar Dataset

As an important part of the Chinese Meridian Project, two MST radar systems are designed and constructed to improve the understanding of the extratropical troposphere, lower stratosphere, and mesosphere (Wang, 2010), which are Wuhan and Beijing MST radars. The Beijing MST radar located in Xianghe, Hebei Province, China (39.75° N, 116.96° E, 22 m above sea level) was designed and constructed by the Institute of Atmospheric Physics, Chinese Academy of Sciences and started its routine operation since 20 October 2011 (Tian and Lu, 2017). The radar is a high power coherent pulse-Doppler radar operating at 50 MHz with the maximum peak power of 172 kW and the half-power beam width of 3.2°. Five beams are applied: one vertically pointed beam and four 15° off-zenith beams tilted to north, east, south, and west. In order to obtain the high-quality measurements from troposphere, lower stratosphere, and mesosphere simultaneously, the radar is designed to operate routinely in three separate modes: low mode (designed range 2.5-~12 km), middle mode (10-~25 km), and high mode (60-~90 km) with vertical resolutions of 150, 600, and 1200 m, respectively. Under the routine operation, the 15-min break is followed by the 15-min operation cycle (5 min for each mode). As a result, the time resolutions of the low, middle, and high mode measurements are all 30 min. More detailed review of the radar system is given by Chen et al. (2016).

Here only the low mode echo power measurements are used to determine the RT height. Although the designed detectable range of the low mode is from 2.5-~12 km, the vertically pointed beam can receive stronger echoes from a higher level (~14-15 km)



as compared with those from off-vertical beams due to the partial specular reflection
mechanism. The measurements in middle mode are also applied to calculate the winds
or echo power within ~5-6 km of the tropopause. The parameters for the two routine
operation modes are listed in Table 1. The monthly total number of the echo power
profiles available in vertical beam (low mode) is shown in Fig. 1. The outliers or
severely contaminated data that mainly induced by system problems are eliminated.
The large data gap in September is due to the annual preventive maintenance.
**2.2. Tropopause Definitions**

Due to the large gradient in potential temperature, radar return power received at

vertical incidence is significantly enhanced upon the transition zone of the tropopause
layer. Using this characteristic, the RT height can be determined effectively by the VHF
radar. Here, the RT is defined as the altitude (above 500 hPa) where the maximum
vertical gradient of echo power is located (Vaughan et al., 1995; Alexander et al., 2012;
Ravindrababu et al., 2014; Chen et al., 2018). Considering the occasional and random
noise, to which the derived-RT is sensitive, the echo power profiles are smoothed by a
3-point running mean. In order to further reduce the influence of the noise, the RT
definition used here need to satisfy an additional criterion: the determined RT height
should be continuous with the adjacent RT heights (one on each side), otherwise to
search for the second peak gradient (eliminated if the second peak does not meet the
additional criterion). The "continuous" here means that the discrepancy between the
two successive heights (in time, 0.5-hour interval) should be <0.6 km. A typical
example of the RT and LRT is illustrated in Fig. 2. The LRT is identified based on the





World Meteorological Organization (WMO) criteria (WMO, 1957). The radar aspect
sensitivity is expressed as the ratio between vertical ($p_v$) and oblique ($p_o$) beam echo
power (here is 15° east beam). The radiosonde soundings are launched twice daily from
the Beijing Meteorological Observatory (39.93 ºN, 116.28 ºE, station number 54511),
which is less than 45 km to the radar site. In this case, the LRT and RT consistent well
and are at 11.65 km and 11.85 km respectively. As expected, the LRT characterized by
a rapid increase in potential temperature gradient also corresponds to the large gradient
in radar aspect sensitivity. Note that the height with maximum value in echo power lie
at a higher altitude (as compared with the RT height) of ~700 m above the LRT. The
dynamical tropopauses used in this paper are derived from the European Centre for
Medium-Range Weather Forecasts (ECMWF) ERA-Interim Reanalysis (Dee et al.,
2011) and defined as the surface of 2 PVU potential vorticity, which is same to that
used by Sprenger et al., (2003) and Alexander et al. (2012).
**2.3. Tropopause sharpness definition**
For the compared data pairs between the RT and LRT, we calculate the
corresponding tropopause sharpness that represents the strength of the tropopause
inversion layer. As defined by Wirth, (2000), the tropopause sharpness S_TP can be
calculated as:
$$S_{TP} = \frac{T_{TP+\Delta z} - T_{TP}}{\Delta z} - \frac{T_{TP} - T_{TP-\Delta z}}{\Delta z} \qquad (1)$$
where TP denotes the tropopause height, $\Delta z$ =1 km, and $T_{TP}$ indicates the
corresponding temperature. This definition is also used in Alexander et al. 2012 and
we're using it for a good comparison with our results.




## 3.  Results

### 3.1.  High-resolution radar tropopause structure

The fine-scale height-time cross section of radar echo power and aspect sensitivity
is shown in Fig. 3 for a typical month (February 2014), along with the RT, PVT and
LRT marked in the figure. In general, the RT agreed well with both the LRT and PVT
in height, and most of the RT exhibit a slightly higher altitude. However, the differences
between the RT and LRT are sometimes large (reach to ~1-2 km) especially when the
RT experience rapid change. Regardless of the background synoptic condition, the
difference in the definitions themselves is to a large degree the main contributing factor
for the large difference between the RT and LRT. For example, a second layer with
significant enhanced echo power is observed above the radar-derived RT for the cases
on 4 and 5 February 2012 (Fig.3a). According to the definitions, the RT well defined as
the first layer with echo enhanced and the LRT matched the second layer, similar to that
observed by Yamamoto et al., (2003) and Fukao et al., (2003). It is of note that the RT
well separates the troposphere characterized by low aspect sensitivity from the lower-
stratosphere characterized by high aspect sensitivity (Fig.3b).

### 3.2.  Comparisons between different definitions

To further quantify the consistency and difference in altitude between different
tropopause definitions, a detailed comparison is carried out in this section. The seasonal
scatterplots for RT versus LRT and the histogram distribution of altitude differences
between the RT and LRT are illustrated in Fig. 4, during the period November 2011-



May 2017. A total of 2411 data pairs are obtained for comparison. Among them, the
number of data pairs is 845 for DJF (winter), 721 for MAM (spring), 321 for JJA
(summer), and 524 for SON (autumn). Comparisons have shown a good consistency
throughout the seasons and most of the RTs exhibit a slightly higher than the LRTs. The
correlation coefficient is 0.74, 0.80, 0.82, and 0.78 for DJF, MAM, JJA, and SON,
respectively. The mean and standard deviation difference (RT minus LRT) calculated
in DJF, MAM, JJA, and SON is $(0.14 \pm 0.75)$, $(0.26 \pm 0.78)$, $(0.33 \pm 0.56)$, and
$(0.12 \pm 0.69)$ km, respectively. The proportion of the data pairs with differences <500 m
is reasonably good during four seasons and is 63%, 61%, 64%, and 67% for DJF, MAM,
JJA, and SON, respectively. Fig. 4 explicitly indicates the good capability of the Beijing
MST radar to determine the tropopause structure well throughout the seasons.

To examine the potential role of the sharpness, Fig. 5a and Fig. 5b show the

histogram distribution of the tropopause sharpness along with the probability density
curve for data pairs with difference (absolute values of RT minus LRT) <0.5 km and >1
km respectively. What is apparent is that most data pairs of Fig. 5a are located to the
right (higher sharpness values, with the peak of ~7.06 K/km) and of Fig. 5b are to the
left (lower sharpness values, with the peak of ~6.35 K/km). No matter whether this
distribution feature is associated with the cyclonic-anticyclonic systems (e.g. Randel et
al., 2007; Randel and Wu, 2010), the results more or less demonstrate that the larger
(weaker) tropopause sharpness contribute to lower (higher) difference between the RT
and LRT. From the perspective of seasonal statistics, the tropopause sharpness over
Beijing station shows similar distribution characteristics throughout the seasons (not





shown), which is different from that in polar regions where the sharpness is significantly
higher during summer than during winter (Zängl and Hoinka, 2001).
The seasonal scatterplots and height difference distribution between the RT and
PVT are illustrated and quantified in Fig. 6. The total number of comparing data pairs
for winter, spring, summer, and autumn is 1422, 1260, 791, and 1145, respectively.
During winter and spring (Fig. 6a and 6b), the RTs agree reasonably well with the PVTs
with the correlation coefficient of 0.72 and 0.76 and the mean difference (RT minus
PVT) of (0.55±0.84 km) and (1±0.89 km), respectively. In contrast, the consistency
for summer and autumn (Fig. 6c and 6d) is relatively bad and with correlation
coefficient of 0.33 and 0.47 and mean difference of (0.80±1.39 km) and (0.75±1.23
km), respectively. Especially for summer, the proportion of the comparing data pairs
with difference <0.5 km is only 10.6% (84). In autumn, need to note that most data pairs
with poor consistency is sampled during early autumn.
**3.3. Observational characteristics in the vicinity of the tropopause**
Measurements of radar middle mode are used for examining the horizontal wind,
return power, and effective data acquisition rate within 5-6 km of the tropopause (upper
troposphere and lower stratosphere). Left panels of Fig. 7 show the vertical scatterplots
of the static stability (represented by the buoyancy frequency squared) as a function of
height relative to the LRT and the right panels show the radar echo power as a function
of height relative to the RT, during two specific years 2012-2013 for extended winter
NDJFM and summer MJJAS seasons. Results clearly demonstrate the sudden jump in
static stability and rapid increase in echo power upon the corresponding tropopause

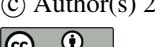



layer. The variation in echo power is more gradual. The amplitude of both the jump and
the increase experienced a slightly larger during NDJFM than that during MJJAS.

Fig. 8 shows the profiles of mean radar effective data acquisition rate for low and

middle modes during November 2011-May 2017. Here, the "effective data" of one
specific range gate requires at least three non-coplanar beams have received
backscattered echoes, by which 3-dimensional wind can be derived. The mean data
acquisition rate profiles both exhibit an obvious inversion layer (i.e. increase
significantly with height) near the tropopause, with the first peak located ~1 km higher
above the mean tropopause height. Note that the second inversion in middle mode
profile that occurred near 16 km is associated with the second tropopause. As limited
by the highest detectable altitude (the data acquisition rate decreased to lower than 20%
at ~16 km), the profile in low mode shows little evidence of second inversion.

Fig. 9 shows time-height intensity plot of the monthly mean radar-derived

horizontal wind (from middle mode) during November 2011-May 2017, together with
the monthly mean location of RT and LRT. One pixel grid denotes 1 month×0.6 km.
The monthly mean RT and LRT agreed well with each other in height, within 400 m in
August and September and even lower in other months of about within 200 m. They
both exhibit a clear seasonal variation, with maximum in early autumn of ~11.6 km and
minimum in early spring of ~10.3 km. The monthly mean wind jet varies with season,
with the thinnest thickness and lowest strength in summer. The mean tropopause height
appears to correspond to the lower boundary location of peak wind layer. The error bars
of both the RT and LRT help to illustrate that the tropopauses changes by larger



amplitude in winter and June than that in other months.
**3.4.   Periodogram analysis of the radar tropopause**

High temporal resolution detection of tropopause by VHF radar have allowed us

to investigate the diurnal or semidiurnal variability of the tropopause. Atmospheric tides
are well known global oscillations contributing to the diurnal variation in temperature
and background winds, which in turn modulate the tropopause height. With the absence
of high resolution temperature measurements, radar-derived winds are combined used
to represent the evidence of diurnal or semidiurnal variation in tropopause height that
modulated by tidal. The frequency power spectrum of the RT height, zonal and
meridional wind, calculated by means of Lomb-Scargle method (Press and Rybicki,
1989), is illustrated in Fig. 10 for two typical months: May 2015 and December 2016.
The choice of Lomb-Scargle algorithm is due to the presence of data gaps (~2 days per
week, especially during 2012-2013). The dominant ~24 h periodicity in RT height,
zonal and meridional wind is obvious for both months. The evidence of ~12 h period in
all three parameters is distinct for May 2015 (Fig. 10a), although the power is relatively
weaker. Through the analysis for each individual month, we found that the semidiurnal
component in the three parameters is generally and occasionally observed in summer
and later spring during our experimental period. The characteristics of the diurnal
variation of the RT height can be represented better in Fig. 11, which shows the mean
Lomb-Scargle power spectrum of the RT as a function month during November 2011-
May 2017. As compared with other months, the dominant diurnal periodicity is less
evident in April. We need to clarify that atmospheric tides are of course not the only



source of the diurnal variation in tropopause height, diurnal convective activities
(Yamamoto et al., 2003) might also be an important cause. Here will not be detailly
discussed.

## 292    **4. Discussion**

As for the radar echo power definition, the RT estimation sometimes will fail due
to the system problems, even if the thermal tropopause is well defined (Hall et al., 2009).
Apart from the system problems, the following two conditions are primarily responsible
for the failure (or difficulty) of both the radar and thermal definitions over the radar site
latitude (~40° N). Firstly, the temperature sometimes continue to decrease upon into the
lower stratosphere (below 16 km) in summer and early autumn, leading to the
failure/difficulty of both the radar and thermal definitions (a typical case as shown in
Fig. 12a). Need to note that the temperature inversion layer occurred at ~16 km in
summer or early autumn is the second tropopause with characteristics of Tropics (Pan
et al., 2004; Randel et al., 2007). Secondly, some specific meteorological processes can
lead to the ambiguities and indefiniteness in thermal and radar definitions, such as
fronts, cyclones or typhoons, and folding (e.g. Nastrom et al., 1989; May et al., 1991;
Roettger, 2001; Alexander et al., 2013). Such ambiguities often result in large difference
in altitude between the RT and LRT. Especially when multiple temperature inversion
layers occurred (below 16 km), the RT generally matched the lower part and LRT often
matched the upper part (e.g. Yamamoto et al., 2003; Fukao et al., 2003), such as the
double layers of enhanced echo power shown in Fig. 3 on 4 and 5 February 2012. Apart



from the two situations above, there is another condition that is commonly responsible
for the failure of thermal definition in summer and early autumn. As the typical case
shown in Fig. 12b, a significant inversion in temperature (at ~12 km) is recorded from
the radiosonde profile, but this inversion layer is too thin and weak to meet the WMO
criterion that thermal definition required. Whereas, the apparent enhancement in radar
echo power corresponding to such inversion layer is strong enough to well define the
RT. Need to highlight again that the temperature inversion layer that occurred near ~16
km is the second tropopause (not considered here). The conditions mentioned above are
the main reasons for fewer comparison data pairs in summer than that in other seasons
(Fig. 4c and Fig. 6c).

Pan et al., (2004) have reported that the difference between the LRT and PVT are

more distinct in the vicinity of subtropical jet. In the northern hemisphere, the axis of
the subtropical jet is situated near ~30°N in spring and winter, whereas in summer and
early autumn the subtropical jet shifts northward to ~40°N (see Fig. 4 in Ding and Wang,
2006). We preliminary considered that the bad consistency between the RT and PVT in
summer and early autumn (Fig. 6c and 6d) is most likely associated with the subtropical
jet shifting poleward to ~40°N. The existing cyclones or anticyclones in the upper-
troposphere (Wirth, 2000), of course, may also be an important cause of the significant
asymmetric differences (scattered points deviate significantly from the 1:1 line and
PVT located below the RT in most cases, as shown in Fig. 6c). More detailed discussion
about the striking asymmetric differences in height between LRT and PVT can be seen
in Wirth (2001). Anyway, we need to be careful when using the dynamical definition to





define the tropopause over radar site latitude ~40° N, especially in summer.

About the characteristics of tropopause and the comparison between different

definitions, there are many differences between mid-latitude and polar regions. In mid-
latitude (~40°N), our results show that: (1) the agreement between RT and LRT is
similar good throughout the seasons; (2) RTs are generally located higher than the LRT;
(3) the thermal definition sometimes fail in summer and early autumn; (4) the
agreement between the RT/LRT and PVT in summer is poor. Whereas, in contrast,
previous researches about the tropopause over polar regions showed that (Wirth, 2000;
Alexander et al., 2012): (1) the difference between the RT and LRT is larger during
winter than that during summer; (2) RTs are generally located lower than the LRT; (3)
the thermal definition sometimes fail in winter and spring; (4) comparison between the
RT and PVT showed the similar good agreement during both summer and winter.

Over a polar latitude station, the seasonal characteristics of the diurnal oscillation

in tropopause height were investigated using 5 years of SOUSY VHF radar
measurements (Hall, 2013b). The sunlight variability in polar regions is different from
that in other latitudes of the world. Different sunlight variation actually will lead to
difference in atmospheric tides, and then would result in different diurnal variation in
tropopause height. Here we found that the diurnal oscillation of RT height at Xianghe
is ubiquitous and obvious throughout the seasons except for April (Fig. 11). Whereas at
polar latitude and in months of November to February when there is no sunlight, Hall
(2013b) observed little evidence of 24 h diurnal variability in RT height.



## 5. Conclusions

In this paper, we present the high resolution structure and variability of the tropopause in Xianghe, China (39.75° N, 116.96° E), based on the Beijing MST radar vertical beam echo power data collected during the period November 2011-May 2017. Fine-scale structure of the RT is well determined with a high temporal resolution of 0.5 h. Comparison results have shown good agreement in altitude between the RT and LRT, with a correlation coefficient of $\geq 0.74$ for the four seasons. Higher tropopause sharpness seems to contribute lower difference between the RT and LRT in altitude and weaker sharpness appears responsible for higher difference. The agreement between the RT and PVT is relatively well in winter and spring with correlation coefficient of 0.72 and 0.76 respectively, but poor during summer with a correlation coefficient of only 0.33. We initially suggested that the poor consistency between RT and PVT is associated with the subtropical jet shifting poleward to ~40°N.

As expected, the sudden jump in static stability (represented by the buoyancy frequency squared) and the rapid increase in radar echo power upon the tropopause layer are clearly observed. A significant inversion (increasing with height) in effective radar data acquisition rate is also observed upon the tropopause layer. Both the monthly mean RT and LRT height have shown a clear annual cycle. The variability and oscillation of RT height with diurnal or lower timescales is presented. Obvious diurnal variation in tropopause height, zonal wind, and meridional wind is generally observed throughout the seasons, indicating the modulation most likely from the atmospheric tides. The semidiurnal variation in RT height is not so obvious and commonly observed



occasionally in summer and late spring.

**Acknowledgment**
This work is funded by National Natural Science Foundation of China (NSFC grants
No. 41474132 and 41722404). We acknowledge the Chinese Meridian Project for
providing the MST radar data. The authors sincerely acknowledge the ECMWF for
providing global reanalysis data. The MST radar data for this paper are available at
Data Centre for Meridian Space Weather Monitoring Project (http://159.226.22.74/).
The radiosonde data are publicly available from the NOAA/ESRL Database at
https://ruc.noaa.gov/raobs/.

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

Contributions of anthropogenic and natural forcing to recent tropopause height
changes. Science, 301(5632), 479-483, 2003.
Santer, B. D., Sausen, R., Wigley, T. M., Boyle, J. S., Achutarao, K., Doutriaux, C.,
Hansen, J. E., Meehl, G. A., Roeckner, E., Ruedy, R., Schmidt, G., and Taylor, K.
E.: Behavior of tropopause height and atmospheric temperature in models,
reanalyses, and observations: Decadal changes. Journal of Geophysical Research,
108(D1), 4002, doi:10.1029/2002JD002258, 2003a.
Sausen, R., and Santer, B. D.: Use of Changes in Tropopause Height to Detect Human



Influences on Climate. Meteorologische Zeitschrift, 12(3), 131-136, 2003.
Seidel, D. J., Ross, R. J., Angell, J. K., and Reid, G. C.: Climatological characteristics
of the tropical tropopause as revealed by radiosondes. Journal of Geophysical
Research, 106(D8), 7857-7878, 2001.
Son, S., Tandon, N. F., and Polvani, L. M.: The fine-scale structure of the global
tropopause derived from cosmic gps radio occultation measurements. Journal of
Geophysical Research Atmospheres, 116, D20113, 2011.
Sprenger, M., Croci Maspoli, M., and Wernli, H.: Tropopause folds and cross-
tropopause exchange: a global investigation based upon ECMWF analyses for the
time period March 2000 to February 2001. Journal of Geophysical Research
Atmospheres, 108(12), 291-302, 2003.
Tian, Y., and Lu, D.: Comparison of Beijing MST Radar and Radiosonde Horizontal
Wind Measurements. Advances in Atmospheric Sciences, 34(1), 39-53. doi:
10.1007 / s00376-016-6129-4, 2017.
Vaughan, G., Howells, A., and Price, J. D.: Use of MST radars to probe the mesoscale
structure of the tropopause. Tellus A, 47(5), 759-765, 1995.
Wang, C.: Development of the Chinese meridian project. Chinese Journal of Space
Science, 30(4), 382–384, 2010.
Wirth, V.: Thermal versus dynamical tropopause in upper-tropospheric balanced flow
anomalies. Quarterly Journal of the Royal Meteorological Society, 126(562), 299-

317, 2000.



Wirth, V.: Cyclone-anticyclone asymmetry concerning the height of the thermal and the

dynamical tropopause. Journal of the Atmospheric Sciences, 58(1), 26-37, 2001.

WMO: Definition of the tropopause. WMO Bull., 6, 136, 1957.
Yamamoto, M., Oyamatsu, M., Horinouchi, T., Hashiguchi, H., and Fukao, S.: High

time resolution determination of the tropical tropopause by the Equatorial

Atmosphere Radar. Geophysical Research Letters, 30(21), 2094, 2003.

Zängl, G., and Hoinka, K. P.: The tropopause in the polar regions. Journal of Climate,

14(2001), 3117-3139, 2001.




**Table**

| Radar parameter | Value |
| --- | --- |
| Transmitted frequency | 50 MHz |
| Antenna array | 24×24 3-element Yagi |
| Antenna gain | 33 dB |
| Transmitter peak power | 172 kW |
| Code | 16-bit complementary |
| No. coherent integrations | 128 (low mode)/64 (mid mode) |
| No. FFT points | 256 |
| No. spectral average | 10 |
| Pulse repetition period | 160 (low mode)/320 (mid mode) μs |
| Half power beam width | $3.2^o$ |
| Pulse length | 1 (low mode)/4 (mid mode) μs |
| Range resolution | 150 (low mode)/600 (mid mode) m |
| Temporal resolution | 30 min |
| Off-zenith angle | $15^o$ |

**Table 1.** Routine operational parameters in low and middle mode for the Beijing MST
radar used in this study.

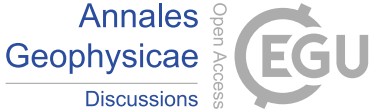



**Figures**

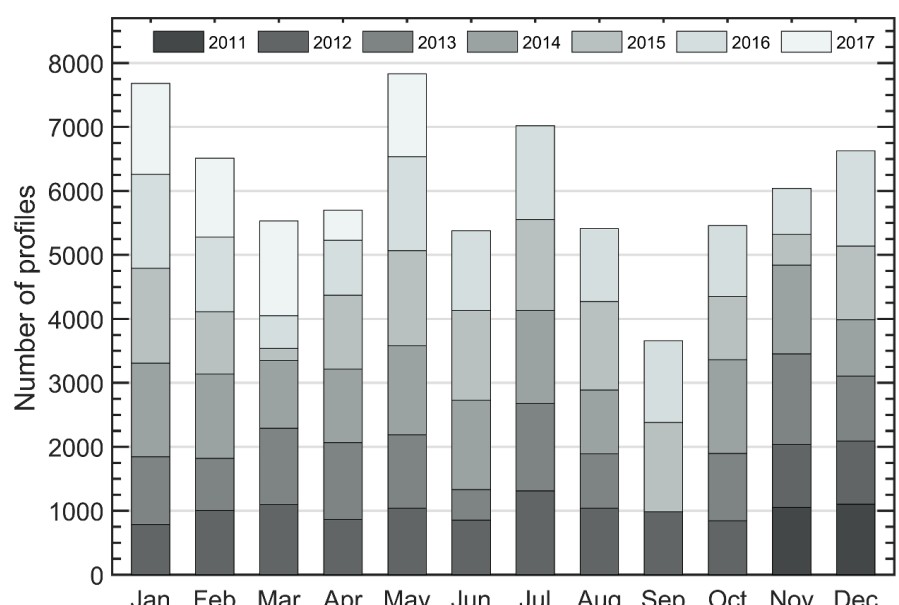

**Figure 1.** Distribution of the monthly total number of radar return echo power profiles

that available from vertical beam in low mode, collected for the period November 2011-

May 2017.



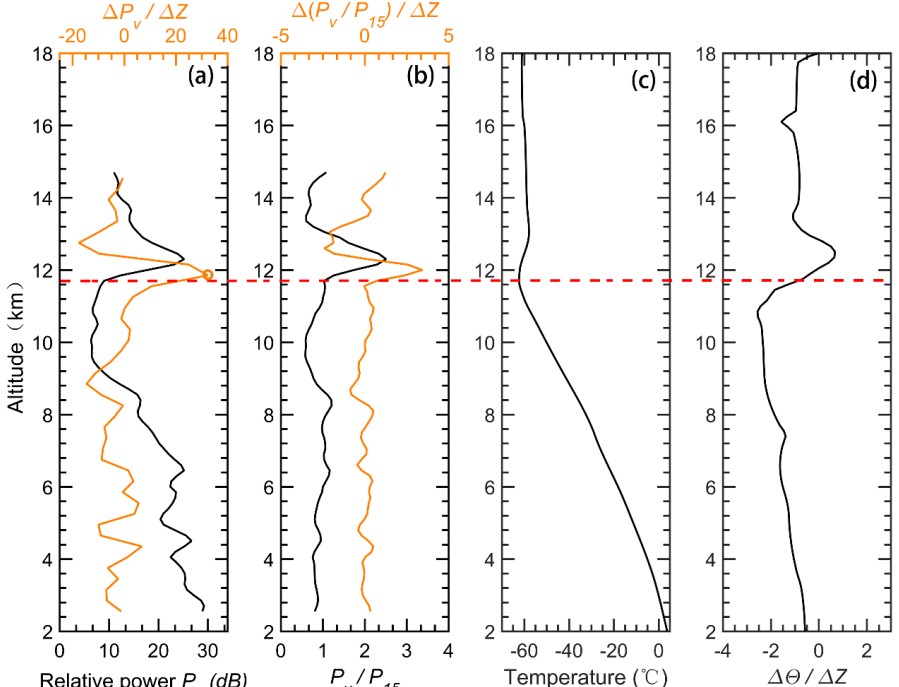

**Figure 2.** Example vertical profiles of (a) relative radar echo power (black line) along

with its gradient variation (orange line), (b) radar aspect sensitivity (black line) along

with its gradient variation (orange line), (c) radiosonde temperature and (d) potential

temperature gradient on 00 UT 04 November 2011. The horizontal red dashed line

marks the LRT height. The orange circle in Fig. 2a denotes the RT height.



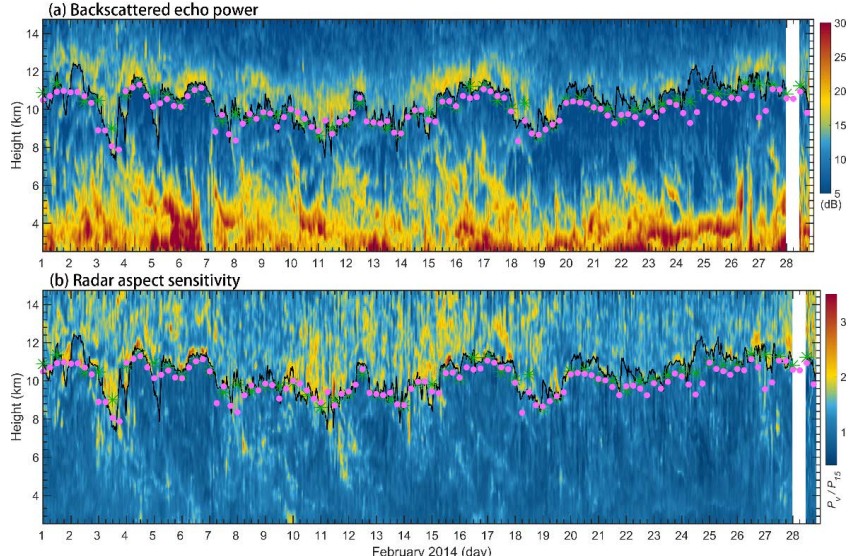

**Figure 3.** Altitude-time intensity plot of (a) radar backscattered echo power and (b) radar aspect sensitivity for February 2014. The tropopauses determined based on the radar echo definition are shown as a black solid curve. The green asterisks '*' and pink dots indicate the location of the LRT derived from simultaneous twice daily radiosonde data and the PVT from ECMWF ERA-Interim reanalysis, respectively. White stripe indicates the time frame of radar missing data.





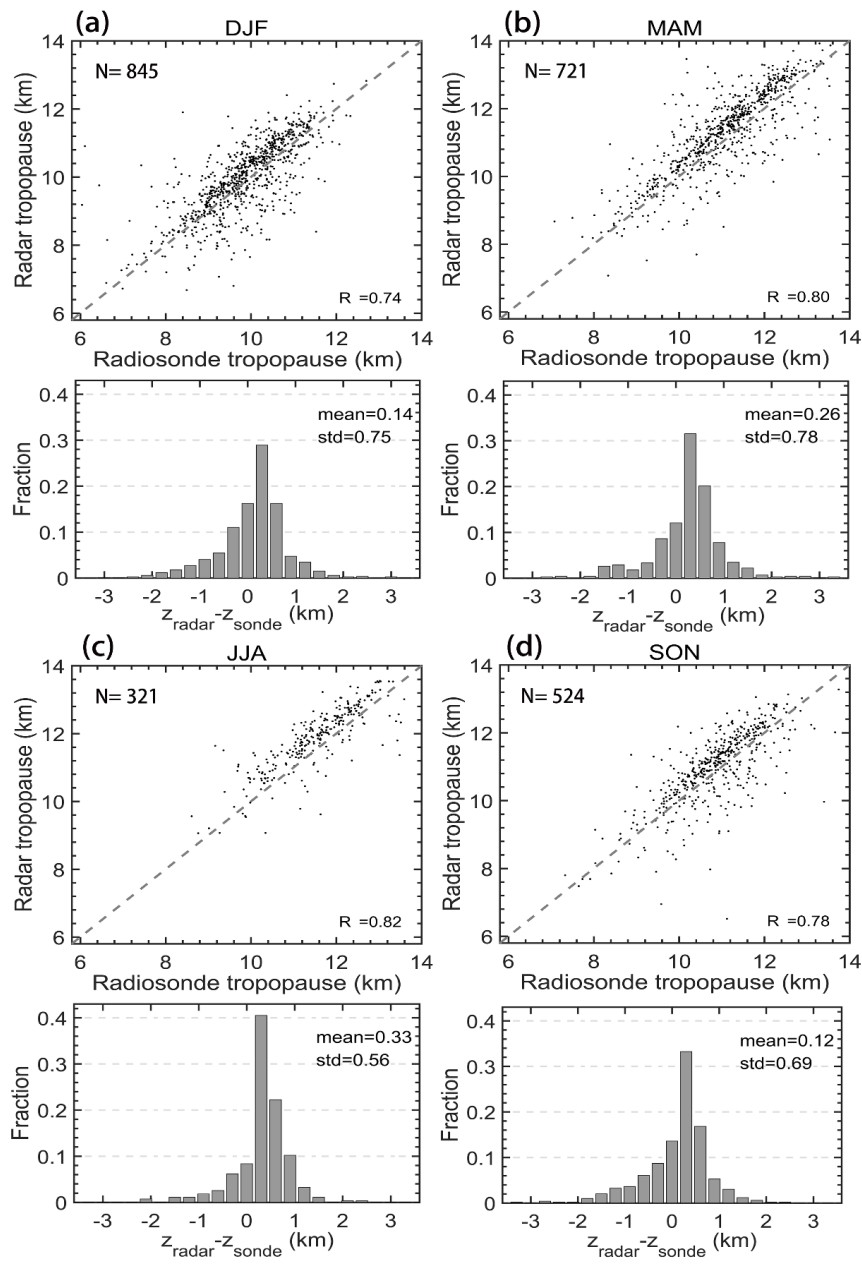

**Figure 4.** Seasonal scatterplots of the RT versus LRT and histogram distribution of

altitude differences between the RT and the LRT, for (a) winter DJF, (b) spring MAM,

(c) summer JJA, and (d) autumn SON, during the period November 2011-May 2017.

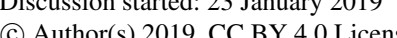

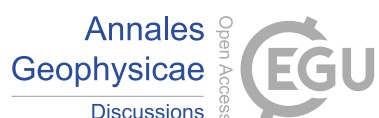

The positive values in the histogram indicate the RT locating at a higher level than the
LRT. The grey dashed line shows the 1:1 line. Here, 'N','$R^2$', 'mean', and 'std' indicate
the sample numbers, correlation coefficient, mean difference, and standard deviation of
the difference, respectively.


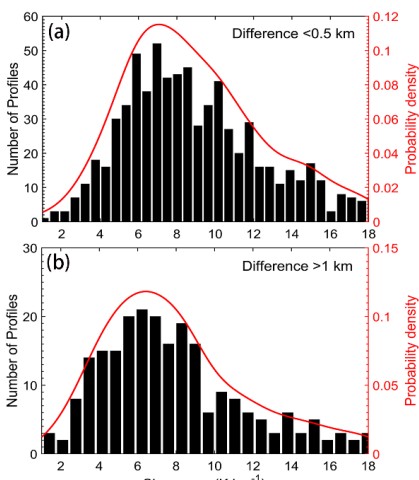


**Figure 5.** Histogram distribution of the tropopause sharpness for (a) difference <0.5
km, and (b) >1 km respectively between the LRT and the RT.





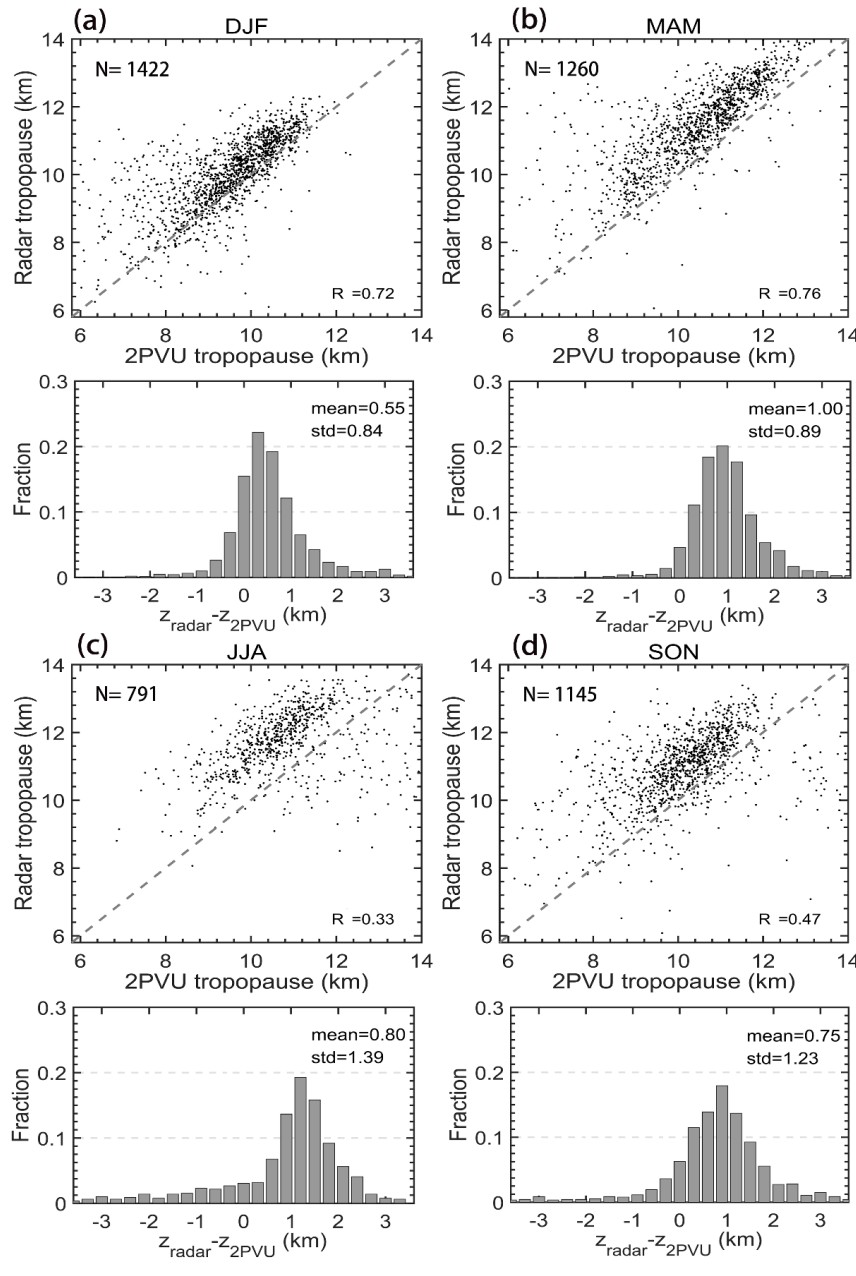


**Figure 6.** Same as figure 4, but for the comparison between the RT and the PVT.


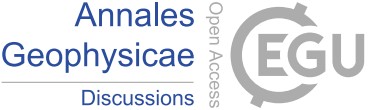

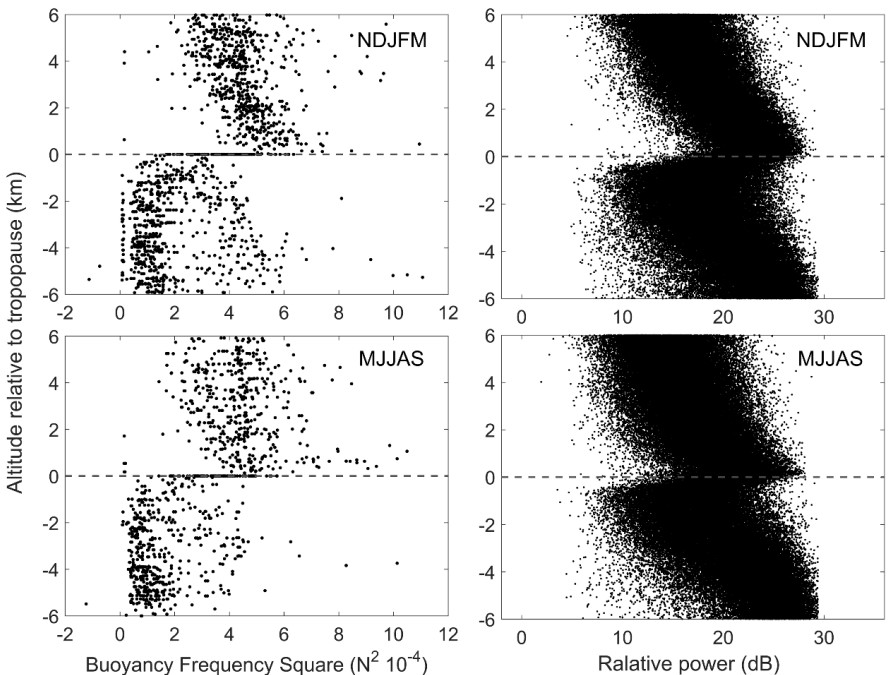


**Figure 7.** Scatterplots of (left panels) static stability ($N^2$) and (right panels) radar

relative echo power as a function of altitude relative to the LRT (left panels) and RT

(right panels) for extended winter (NDJFM) and summer (MJJAS) seasons for two

specific years 2012-2013.





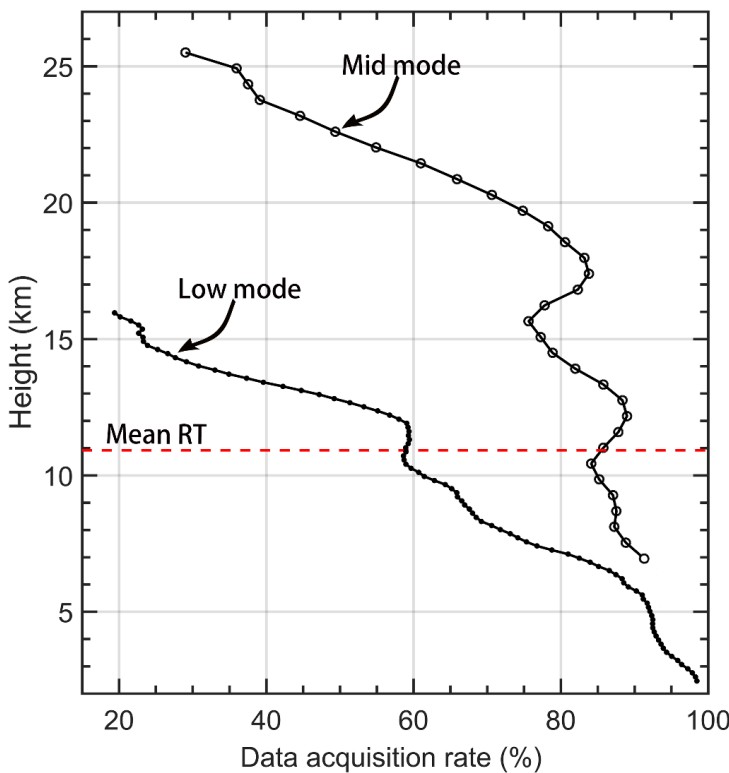


**Figure 8.** Vertical height profiles of the averaged effective radar data acquisition rate

in low mode and middle mode during November 2011-May 2017. The red dashed line

indicates the mean RT height.




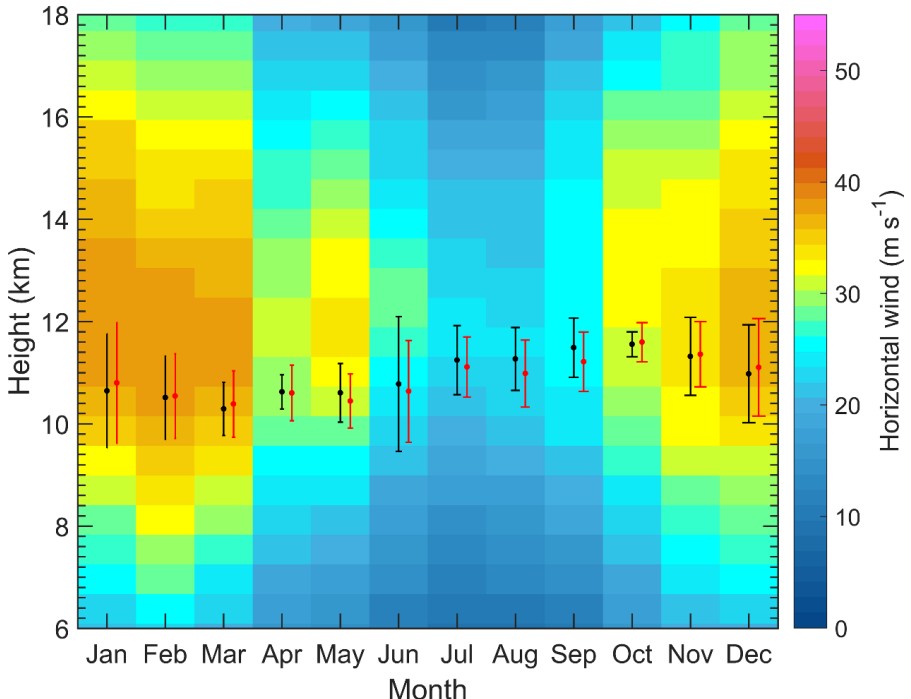

**Figure 9.** Height-time intensity map of monthly mean horizontal wind speed (shaded,

m/s) derived from the middle mode of Beijing MST radar, during November 2011-May

2017. Also shown is the monthly mean height of RT (black dots) and LRT (red dots,

offset by +6 days) along with the vertical error bars representing the standard deviations.

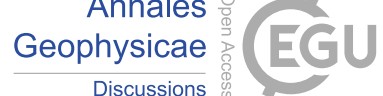

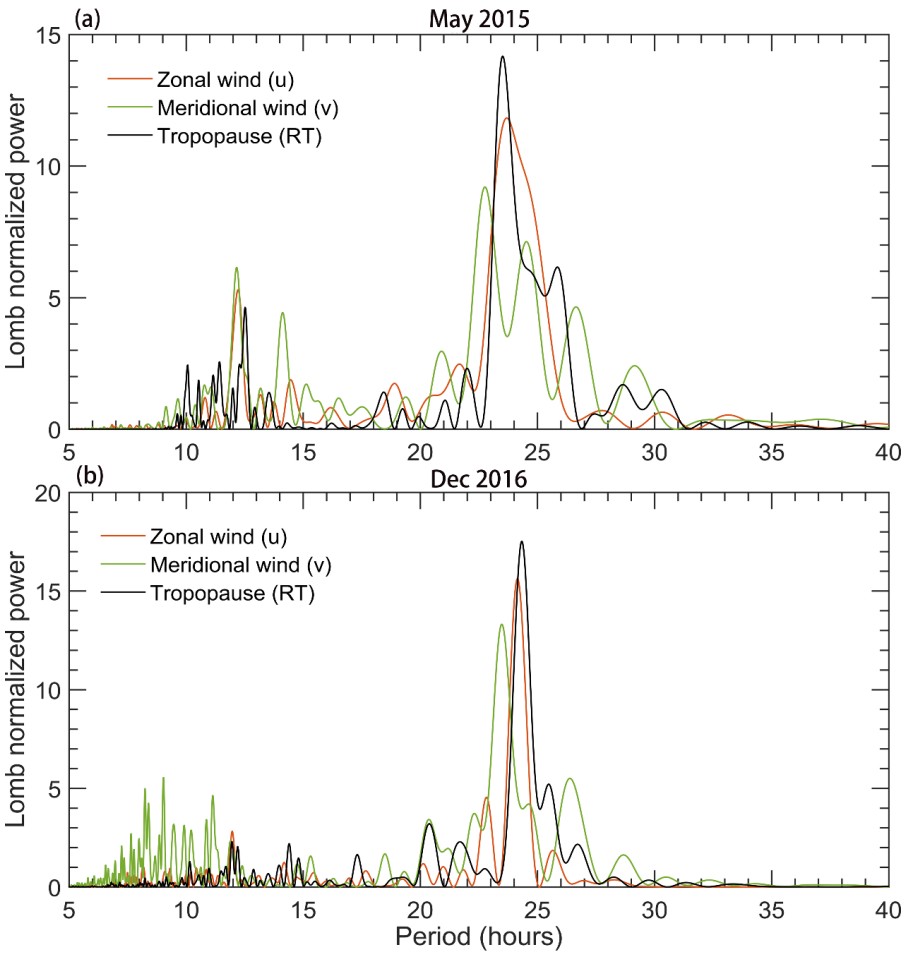

**Figure 10.** Lomb-Scargle periodograms of the RT height, zonal, and meridional wind

oscillations for specific months of (a) May 2015 and (b) December 2016. The zonal and

meridional wind for (a) is sampled at 9.85 km and (b) at 11 km.

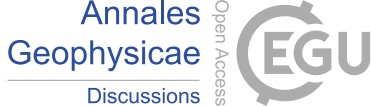



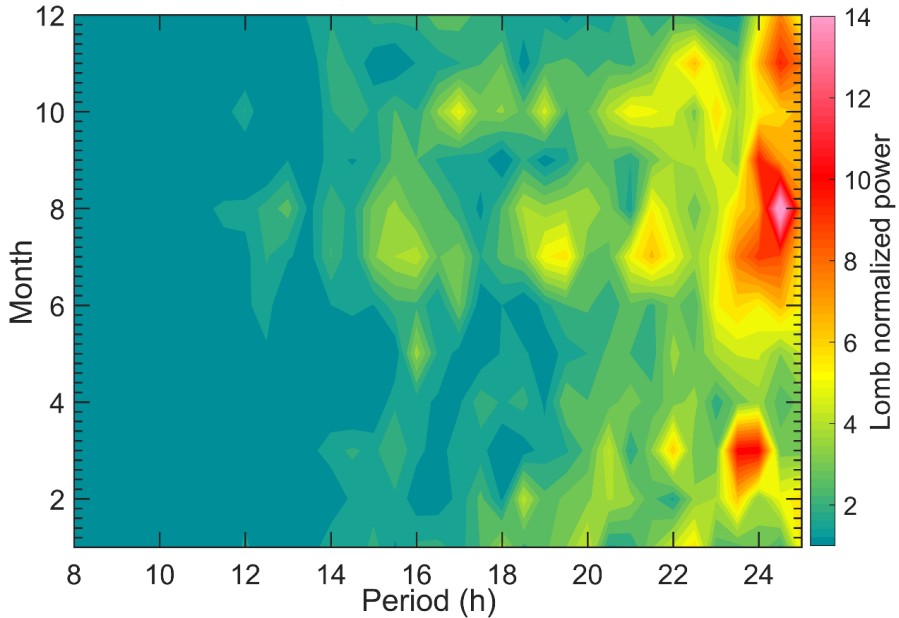

602

**Figure 11.** Mean Lomb-Scargle periodograms of RT height as a function of the time of

month during November 2011-May 2017.

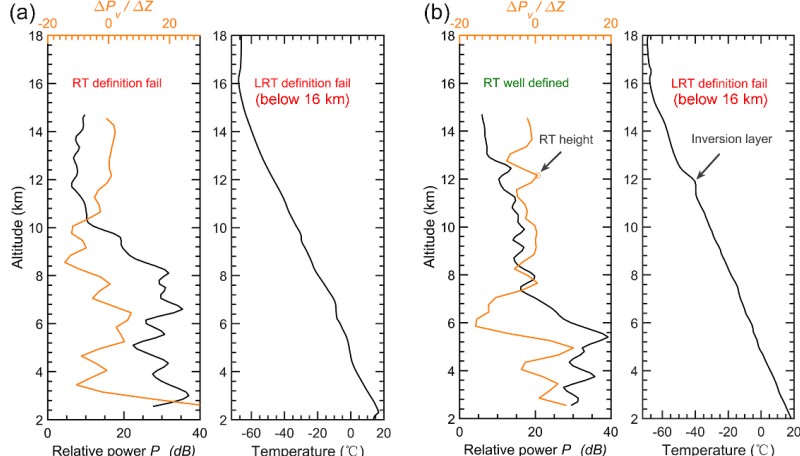

605

**Figure 12.** Example profiles of radar echo power and radiosonde temperature that (a)

both the RT and LRT definitions fail due to the continuing decrease in temperature on

00 UTC 7 July 2012 and (b) the temperature inversion layer failed to meet the LRT

definition but well defined in RT definition on 12 UTC 02 August 2012.