# Peer review of "High-resolution Beijing MST radar detection of tropopause structure and"

_Annales Geophysicae, 2018_

## Referee Comment (RC1) · Anonymous Referee #1 · 21 Feb 2019

Summary

Chen and co-authors use a VHF wind-profiling radar to examine the structure and variability of the tropopause over Xianghe, China with high temporal resolution. The authors use the gradient of the return power to identify the radar tropopause (RT) and compare with lapse-rate tropopauses (LRT) calculated from radiosondes which are launched some 45km away. The RT and LRT agree fairly well: the authors speculate that the non-perfect correlations are due to the seasonal movement of the sub-tropical jet. Chen and co-authors also investigate the tropopause sharpness and the relation between RT and the 2PVU dynamical tropopause and briefly mention the syn-

optic meteorology likely contributing to the similarities and differences between these tropopause definitions.

As Chen and co-authors cite in their manuscript, there have been several papers discussing the structure of the RT at varying latitudes in recent years. It seems that there is a bit of a renaissance in papers on this subject following the early study of Gage & Green (1979). The ability of VHF radars to sample sub-diurnal tropopause structure still must be of interest to the community.

Furthermore, the latitude of Xianghe at 40N is interesting for sub-diurnal and synoptic-scale tropopause variability given the seasonal movement of the sub-tropical jet often creating two tropopauses which allows for significant stratosphere-troposphere exchange. Unfortunately, Chen and colleagues choose to reject any analysis of the second, higher, tropopause in this paper which would have added significant interest to their analysis. As such, as it stands, this paper contains little new or interesting science.

I would therefore like the authors to implement the following recommendations in order to increase the scientific value and interest of their study.

Paper references are given at the end.

Major Comments

1) Line 101. You write that you will focus on the first (lowest) tropopause here. Yet I would argue that by neglecting the second tropopause, you are majorly limiting the value of your science. For example, by also characterising the second tropopause, you would likely answer some of your speculations regarding the differences between RT, LRT which you make in the Conclusions. Examining the seasonal variations of both tropopauses with radar would be a useful contribution to the literature and should be done in this paper.

2) Line 230 regarding the low correlations during summer and autumn. Although the RT

and PV tropopauses are both dynamical tropopause definitions, we would not always expect close agreement especially during the passage of cyclones. Still, these larger offsets during summer (Fig 6c) are interesting. Does this suggest that the 2PVU surface is not the best measure of a dynamical tropopause above Beijing during summertime? Given that you only consider 'low-mode' tropopauses, maybe you are missing most of the summer high tropopauses (see Major Point 3 below) which may account for some differences? You should separate the tropopause data into cyclonic / anticyclonic conditions, as you should then discover the reasons for the difference – my expectations is that you should see closer agreement between the definitions during anti-cyclonic conditions than during cyclonic. Also, you should separate the data into single tropopause / double tropopause times to investigate the RT – PV tropopause relationships more fully.

3) Discussion section and Figure 12a. I am surprised that you can't detect the thermal (lapse-rate) tropopause in the radiosonde profile at 16km. In such cases, as your radar 'low mode' doesn't reach high enough, you should switch to analysing 'mid-mode' to find the radar tropopause. On line 300 you say that the 16km inversion is the 'second tropopause' but clearly it's the first tropopause, because it is the lowest altitude tropopause. Presumably on this day, you are observing a tropical-like atmosphere with a very high tropopause. You should be aware that at a similar latitude to the Beijing radar is the MU radar in Shigaraki, Japan (35N), where high-time resolution radiosonde and radar analyses over many decades have demonstrated the very high summertime radar and radiosonde tropopauses (first tropopause) at altitudes above 15km (please refer to Tsuda et al., 1991; Hermawan et al., EPS, 1998; Alexander & Tsuda, JTech, 2008, for further details). You need to include analysis of your higher-altitude tropopause in your study, regardless of whether it's the first or second tropopause.

Minor Comments 1) Line 51: 'Radiosonde sounding… impractical [spelling!] in severe weather'. This isn't really true. The sentence suggests to the reader that radiosondes cannot be launched in heavy rainfall yet they are in the tropics, nor in the cold, yet they

are in the polar regions. I suggest removing this sentence.

2) Line 62. So the best way forward is to create a 'blended tropopause' for the globe as Wilcox et al. (QJRMS 2011) did. I suggest you read and cite this paper here.

3) Line 63. Before discussing VHF radars, you should briefly discuss the use of GPS radio occultation satellites which provide highly accurate, climatically stable measurements of temperature and thus of the tropopause. There are many papers on this subject which you can easily find. Some valuable ones include: Schmidt et al., ACP 2005; Son et al., JGR 2011.

4) Line 71. I cannot find the paper Alexander et al (2012). I think you are accidently quoting the ACPD submitted manuscript paper rather than the final ACP paper. You should cite the final, ACP paper as: Alexander et al. (2013). See the 'References' section below for the proper reference.

5) Line 102. For what purpose is the high temporal resolution 'still insufficient'? Why do we care about obtaining the tropopause at hourly time-scales?

6) Line 209, sentence: 'Fig 4 explicitly indicates the good capability of the Beijing MST radar...' No it doesn't. Figure 4 shows that the radar tropopause determined by this radar shows reasonable agreement with radiosonde-derived lapse rate tropopauses and that the differences are mostly under 1km.

7) Line 245: I can't see these differences clearly in Figure 7. On top of your dot points in Figure 7, please plot the mean and standard deviations at each altitude. Please describe in the text what new information this plot shows – after all, the community well knows about the sudden jump in $N^2$ at the tropopause and the jump in radar power too.

8) Line 312 onwards (comments about Figure 12b and the inversion at 12km). A simple way around this is to set some threshold in your radar tropopause algorithm to avoid these small peaks. And again, you should switch to 'mid-mode' analysis here.

9) Line 326 and comments about cyclones / anti-cyclones. This is easy to investigate and should be done rather than just speculating. See my Major Comments above.

Technical:

1) Line 250: Avoid using the word 'inversion' unless referring to the increase in temperature with altitude

2) Figure 3. I can't see the green asterisks. Please choose a different colour to make this clear

3) Figure 7. The means (and standard deviations) should be overplotted

4) Figure 8. Please clarify the x-axis 'Data acquisition rate'. Is this the percentage of (useful) signal returned, or is it the percentage of wind data collected or what?

5) Figure 12, as discussed in the Minor Comments above, you should be switching to 'mid-mode' to identify the high-altitude tropopause, which in these instances is still the 'first tropopause'

References:

Alexander, S.P., Murphy, D.J., and Klekociuk, A.R., 2013, High resolution VHF radar measurements of tropopause structure and variability at Davis, Antarctica (69° S, 78° E), Atmos. Chem. Phys., 13, 3121-3132, doi:10.5194/acp-13-3121-2013

Alexander, S. P. and Tsuda T., 2008, 'High Resolution Radio Acoustic Sounding System (RASS) Observations and Analysis up to 20km', Journal of Atmospheric and Oceanic Technology, 25, 8, p1383-1396, doi: 10.1175/2007JTECHA983.1

Hermawan, E., and T. Tsuda, 1999: Estimation of turbulence energy dissipation rate and vertical eddy diffusivity with the MU radar RASS. J. Atmos. Solar-Terr. Phys., 61, 1123–1130.

Schmidt, T., Heise, S., Wickert, J., Beyerle, G., Reigber, C., 2005, GPS radio occultation with CHAMP and SAC-C: Global monitoring of thermal tropopause parameters, Atmos. Chem. Phys, 5, 1473—1488

Son, S.-W., Tandon, N.F., Polvani, L.M, 2011, The fine-scale structure of the global tropopause derived from COSMIC GPS radio occultation measurements, J. Geophys. Res, 116, D20113, doi: 10.1029/2011JD016030

Tsuda, T., T. E. VanZandt, M. Mizumoto, S. Kato, and S. Fukao, Spectral analysis of temperature and Brunt Väisälä frequency fluctuations observed by radiosondes, J. Geophys. Res., 96, 17265–17278, 1991.

Wilcox L.J., Hoskins B.J., Shine K.P. 2012. A global blended tropopause based on ERA data. Part I: Climatology. Q. J. R. Meteorol. Soc. 138: 561–575. DOI:10.1002/qj.951

---

## Author Comment (AC1) · 24 Feb 2019

Summary

Chen and co-authors use a VHF wind-profiling radar to examine the structure and variability of the tropopause over Xianghe, China with high temporal resolution. The authors use the gradient of the return power to identify the radar tropopause (RT) and compare with lapse-rate tropopauses (LRT) calculated from radiosondes which are launched some 45km away. The RT and LRT agree fairly well: the authors speculate that the non-perfect correlations are due to the seasonal movement of the sub-tropical jet. Chen and co-authors also investigate the tropopause sharpness and the relation between RT and the 2PVU dynamical tropopause and briefly mention the synoptic meteorology likely contributing to the similarities and differences between these tropopause definitions.

As Chen and co-authors cite in their manuscript, there have been several papers dis-cussing the structure of the RT at varying latitudes in recent years. It seems that there is a bit of a renaissance in papers on this subject following the early study of Gage & Green (1979). The ability of VHF radars to sample sub-diurnal tropopause structure still must be of interest to the community.

Furthermore, the latitude of Xianghe at 40N is interesting for sub-diurnal and synoptic-scale tropopause variability given the seasonal movement of the sub-tropical jet of-ten creating two tropopauses which allows for significant stratosphere-troposphere ex-change. Unfortunately, Chen and colleagues choose to reject any analysis of the second, higher, tropopause in this paper which would have added significant interest to their analysis. As such, as it stands, this paper contains little new or interesting science.

I would therefore like the authors to implement the following recommendations in order to increase the scientific value and interest of their study.

Paper references are given at the end.

**Response:**

We really thank you for the helpful and constructive comments, which will be of great useful for this article. We hope that the reviewers will be satisfied with our responses and revisions. Our responses are in different color style (reviewer's comments are shown in black and our response in blue type).

Response regarding to the comment: 'The RT and LRT agree fairly well: the authors speculate that the non-perfect correlations are due to the seasonal movement of the sub-tropical jet'. Statistically, our results found that the agreement between the RT and LRT height is similarly well during different seasons. We speculate that the relatively poor agreement between the RT/LRT and PVT in summer and late autumn is probably due to the seasonal movement of the sub-tropical jet.

Regarding the second tropopause, detailed responses are given below.

Regarding the scientific value:

Firstly, this paper used the latest data set of Beijing MST radar (more than 5 years since the routine operation of the radar) to study the high-resolution tropopause structure over Xianghe and then compared it with LRT and PVT. The results of this paper are of great guiding significance to readers who want to make use of the Beijing MST data to study various interesting topics (especially the tropopause variation).

Secondly, there are few statistical studies on the tropopause structure at near 40N with high temporal resolution. By comparing with LRT, we verified the potential of Beijing MST radar to identify tropopause. Diurnal variations of the tropopause with high temporal resolution are also analyzed. The echo power intensity, wind field intensity and wind data acquisition rate near the tropopause are also analyzed.

Major Comments

1) Line 101. You write that you will focus on the first (lowest) tropopause here. Yet I would argue that by neglecting the second tropopause, you are majorly limiting the value of your science. For example, by also characterising the second tropopause, you would likely answer some of your speculations regarding the differences between RT, LRT which you make in the Conclusions. Examining the seasonal variations of both tropopauses with radar would be a useful contribution to the literature and should be done in this paper.

**Response:**

Dear reviewer, the identification and observation of the second tropopause (characterized by tropical features and located near 16 km) is not considered by both the RT definition and LRT definition. The second tropopause (if it existed) can be well detected by radiosonde soundings. However, the low mode observations of Beijing MST radar have a limited highest detectible altitude of ~13-14 km (in vertical direction), thus the routine second tropopause is impossible to be detected under low mode observation. The middle mode observations of Beijing MST radar can reach as high as 24 km, but its altitude resolution is relatively poor with value of 600 m, while the resolution in low mode is 150 m. Thus, the middle mode data is not appropriate to be used to detect high resolution tropopause structure, otherwise will lead to a large error by the limited altitude resolution.

Given that we focused on the first tropopause structure using both the RT and LRT definitions, some responses are needed regarding the differences between RT and LRT. The second tropopause structure may hardly an important factor causing the differences between RT and LRT. As mentioned in the manuscript (discussion section) that some specific meteorological processes can lead to the ambiguities and indefiniteness in thermal and radar definitions, such as fronts, cyclones or typhoons, and folding. Such ambiguities often result in large difference in altitude between the RT and LRT. In addition, when multiple temperature inversion layers (sometimes can be called as multiple tropopauses, sometimes can not, depending on if the inversion layers meet the

WMO LRT definition) occurred below 16 km, the RT generally matched the lower part and LRT often matched the upper part, such as the double layers of enhanced echo power shown in Figure 3 on 4 and 5 February 2012.

2)    Line 230 regarding the low correlations during summer and autumn. Although the RT and PV tropopauses are both dynamical tropopause definitions, we would not always expect close agreement especially during the passage of cyclones. Still, these larger offsets during summer (Fig 6c) are interesting. Does this suggest that the 2PVU sur-face is not the best measure of a dynamical tropopause above Beijing during summer-time? Given that you only consider 'low-mode' tropopauses, maybe you are missing most of the summer high tropopauses (see Major Point 3 below) which may account for some differences? You should separate the tropopause data into cyclonic / anti-cyclonic conditions, as you should then discover the reasons for the difference – my expectations is that you should see closer agreement between the definitions during anti-cyclonic conditions than during cyclonic. Also, you should separate the data into single tropopause / double tropopause times to investigate the RT – PV tropopause relationships more fully.

**Response:**
Yes, these larger offsets during summer (Fig 6c) are probable suggest that the 2PVU sur-face is not the best measure of a dynamical tropopause above Beijing during summer-time. We consider that these differences are less related to the missing of summer high tropopause (second tropopause near ~16 km).
Firstly, during autumn and summer, most of the comparison data pairs located in the left-side of 1:1 line (Fig. 6c and 6d), indicating most of the RT are located higher than the 2PVU tropopause height.
Secondly, if the differences are closely associated to the missing of second tropopauses, the distribution of the scatter points in Figure 6c should be that: most of the comparison data pairs located in the right-side of 1:1 line.

Based on the comments and response above, we have added the following sentences in the revised manuscript (discussion section):
'The existing cyclones or anticyclones in the upper-troposphere (Wirth, 2000), of course, may also be an important cause of the significant asymmetric differences (most of the scattered points deviate significantly from the 1:1 line). This asymmetric differences, that is most of the RT are located higher than the 2PVU tropopause height, suggest that the 2PVU surface is not the best measure of a dynamical tropopause over Beijing during summer-time.'.

Certainly, cyclonic and anti-cyclonic conditions may also be an important influence factor for the differences between RT and PVT (Wirth, 2001). More detailed discussion about the striking asymmetric differences in height between LRT/RT and PVT will not be given in this paper.

*Wirth, V.: Thermal versus dynamical tropopause in upper-tropospheric balanced flow anomalies. Quarterly Journal of the Royal Meteorological Society, 126(562), 299-317,*

3) Discussion section and Figure 12a. I am surprised that you can't detect the thermal (lapse-rate) tropopause in the radiosonde profile at 16km. In such cases, as your radar 'low mode' doesn't reach high enough, you should switch to analysing 'mid-mode' to find the radar tropopause. On line 300 you say that the 16km inversion is the 'sec-ond tropopause' but clearly it's the first tropopause, because it is the lowest altitude tropopause. Presumably on this day, you are observing a tropical-like atmosphere with a very high tropopause. You should be aware that at a similar latitude to the Beijing radar is the MU radar in Shigaraki, Japan (35N), where high-time resolution radiosonde and radar analyses over many decades have demonstrated the very high summer-time radar and radiosonde tropopauses (first tropopause) at altitudes above 15km (please refer to Tsuda et al., 1991; Hermawan et al., EPS, 1998; Alexander & Tsuda, JTech, 2008, for further details). You need to include analysis of your higher-altitude tropopause in your study, regardless of whether it's the first or second tropopause.

**Response:**

We thought about this a lot during this statistical study.

The inversion layer near 16 km or higher is indeed meet the LRT definition. However, considering that the radar station is located at the middle latitude of 40N and the mechanism of formation of the second tropopause (Pan et al., 2004; Randel et al., 2007; Pan et al., 2009), the inversion height at a height of ~16km (or higher) over the radar station is the second tropopause with tropical characteristics. In fact, the routine occurred second tropopause is almost located near 16km altitude throughout the seasons. In a word, no matter whether the inversion layer at ~16 km is the first tropopause or the second tropopause, such tropical featured higher tropopause will not be considered and studied here.

Therefore, we explain in the introduction that this study only focuses on the first tropopause below 16km, no matter whether it exists or not. Indeed, the routine presented higher tropopause (second tropopause near 16km) in different seasons throughout the year is worthwhile for studying. In view of the limitation of the altitude resolution of the middle mode data in the Beijing MST radar (with value of 600m), it was not used to study the tropical featured second tropopause near 16km, especially for the statistical study. The case observation of the second tropopause near 16km using the meddle mode is worthy of future study.

In order to avoid misguidance and to fit in with the main research focus of this paper, we have indicated in many places that the research focus of this paper is the first tropopause under 16km (as long as it exists).

For example, one sentence has been modified in the introduction section of the revised manuscript: 'In the present study, we focus only on the first tropopause (below 16 km) which will be referred to as 'tropopause' hereafter'.

In addition, the figure 12 and the figure caption have also been modified accordingly:

[Figure]

**Figure 12.** Example profiles of radar echo power and radiosonde temperature that (a) both the RT and LRT definitions fail due to the continuing decrease in temperature on 00 UTC 7 July 2012 and (b) the temperature inversion layer failed to meet the LRT definition but well defined in RT definition on 12 UTC 02 August 2012. Please note that we only consider the conditions below 16 km.

Pan, L. L., Randel, W. J., Gary, B. L., Mahoney, M. J., and Hintsa, E. J.: Definitions and sharpness of the extratropical tropopause: A trace gas perspective. Journal of Geophysical Research, 109, D23103, doi:10.1029/2004JD004982, 2004.

Pan, L. L., W. J. Randel, J. C. Gille, W. D. Hall, B. Nardi, S. Massie, V. Yudin, R. Khosravi, P. Konopka, and D. Tarasick: Tropospheric intrusions associated with the secondary tropopause, Journal of Geophysical Research, 114, D10302, 2009.

Randel, W. J., Seidel, D. J., and Pan, L. L.: Observational characteristics of double tropopauses. Journal of Geophysical Research, 112, D07309, 2007.

Minor Comments

1) Line 51: 'Radiosonde sounding. . . impractical [spelling!] in severe weather'. This isn't really true. The sentence suggests to the reader that radiosondes cannot be launched in heavy rainfall yet they are in the tropics, nor in the cold, yet they are in the polar regions. I suggest removing this sentence.

**Response:**
Really thanks for pointing out the flaw. The corresponding sentence has been removed.

2) Line 62. So the best way forward is to create a 'blended tropopause' for the globe as Wilcox et al. (QJRMS 2011) did. I suggest you read and cite this paper here.

**Response:**
Really thanks for recommending this valuable paper. This paper has been cited in the revised manuscript. Following sentence has been added in the revised manuscript: 'Creating a 'blended tropopause' for the globe may probable a good way forward

(Wilcox et al., 2011).'

3)    Line 63. Before discussing VHF radars, you should briefly discuss the use of GPS radio occultation satellites which provide highly accurate, climatically stable measure-ments of temperature and thus of the tropopause. There are many papers on this subject which you can easily find. Some valuable ones include: Schmidt et al., ACP 2005; Son et al., JGR 2011.

**Response:**
Yes, it is necessary to briefly discuss the use of GPS radio occultation satellites to study the tropopause. The corresponding references have been cited in the revised manuscript. Following sentence is added in the revised manuscript: 'In addition, the data of GPS radio occultation satellites is also an effective way and commonly applied to study tropopause (e.g. Schmidt et al., 2005; Son et al., 2011).'

4)    Line 71. I cannot find the paper Alexander et al (2012). I think you are accidently quoting the ACPD submitted manuscript paper rather than the final ACP paper. You should cite the final, ACP paper as: Alexander et al. (2013). See the 'References' section below for the proper reference.

**Response:**
Really thanks for pointing out the flaw. It has been corrected in the revised manuscript.

5)    Line 102. For what purpose is the high temporal resolution 'still insufficient'? Why do we care about obtaining the tropopause at hourly time-scales?

**Response:**
Yes, the expression of this sentence is inaccurate. We have deleted the sentence in the revised manuscript.

6)    Line 209, sentence: 'Fig 4 explicitly indicates the good capability of the Beijing MST radar. . .' No it doesn't. Figure 4 shows that the radar tropopause determined by this radar shows reasonable agreement with radiosonde-derived lapse rate tropopauses and that the differences are mostly under 1km.

**Response:**
Yes, the expression of this sentence is inaccurate. The corresponding sentence has been changed to 'Fig. 4 explicitly shows that the RT derived by the Beijing MST radar agrees reasonably well with the LRT throughout the seasons.' in the revised manuscript.

7)    Line 245: I can't see these differences clearly in Figure 7. On top of your dot points in Figure 7, please plot the mean and standard deviations at each altitude. Please describe in the text what new information this plot shows – after all, the community well knows about the sudden jump in N^2 at the tropopause and the jump in radar power too.

**Response:**
Yes, you are right. It is necessary to plot the mean and standard deviations in each panel of Figure 7. Really thanks for your comments. The mean values and error bars are plotted in Figure 7 in the revised manuscript.

[Figure]

Indeed, the sudden jump in both the static stability and radar echo power upon the tropopause is commonly well known. Several other features seen from the figure have been reported in the revised manuscript: 'The degree of sudden increase in echo power is more gradual than that in static stability. The amplitude of the sudden increase in radar power experienced a slightly larger during NDJFM than that during MJJAS (red lines of right panels). Another interesting feature in the lower-stratosphere is that both the static stability and radar power points show less disperse during NDJFM than that during MJJAS.'.

8)    Line 312 onwards (comments about Figure 12b and the inversion at 12km). A simple way around this is to set some threshold in your radar tropopause algorithm to avoid these small peaks. And again, you should switch to 'mid-mode' analysis here.

**Response:**
Because the radar echo power is associated with various situations. Threshold may not be appropriate for Beijing MST radar. Furthermore, the phenomenal of enhanced gradient in radar echo power is real, and it just correspond well to the relatively weak inversion of radiosonde temperature near 12 km.

9) Line 326 and comments about cyclones / anti-cyclones. This is easy to investigate and should be done rather than just speculating. See my Major Comments above.

**Response:**
Dear reviewer, really thanks for your comments. Certainly, cyclonic and anti-cyclonic conditions are interesting topics. But it will not be studied in detail in this paper and

this is beyond the scope of this article.

Technical:
1) Line 250: Avoid using the word 'inversion' unless referring to the increase in temperature with altitude

**Response:**
Yes, you are right. The corresponding sentence has been modified to 'Clearly, both profiles exhibit a sudden increase with height near the tropopause' in the revised manuscript.

2) Figure 3. I can't see the green asterisks. Please choose a different colour to make this clear

**Response:**
Really thanks for pointing out the flaw. Fig.3 has been corrected in the revised manuscript.

3) Figure 7. The means (and standard deviations) should be overplotted

**Response:**
Really thanks for pointing out the flaw. The means and standard deviations are plotted in each panel of Figure 7.

4) Figure 8. Please clarify the x-axis 'Data acquisition rate'. Is this the percentage of (useful) signal returned, or is it the percentage of wind data collected or what?

**Response:**
Really thanks for your comments. Data acquisition rate indicates the effective wind data. It has been corrected in the 3.3 section and the figure caption in the revised paper.

5) Figure 12, as discussed in the Minor Comments above, you should be switching to 'mid-mode' to identify the high-altitude tropopause, which in these instances is still the 'first tropopause'

**Response:**
As mentioned above and explained in many parts of the article, we only focus on the tropopause below 16km. The tropopause near 16 km or above is not subject to consideration (statistical analysis). For example, one sentence in the introduction section of the revised manuscript: 'In the present study, we focus only on the first tropopause (below 16 km) which will be referred to as 'tropopause' hereafter'.

References:
Alexander, S.P., Murphy, D.J., and Klekociuk, A.R., 2013, High resolution VHF radar measurements of tropopause structure and variability at Davis, Antarctica (69∘ S, 78∘ E), Atmos. Chem. Phys., 13, 3121-3132, doi:10.5194/acp-13-3121-2013
Alexander, S. P. and Tsuda T., 2008, 'High Resolution Radio Acoustic Sounding System (RASS) Observations and Analysis up to 20km', Journal of Atmospheric and Oceanic Technology, 25, 8, p1383-1396, doi: 10.1175/2007JTECHA983.1

Hermawan, E., and T. Tsuda, 1999: Estimation of turbulence energy dissipation rate and vertical eddy diffusivity with the MU radar RASS. J. Atmos. Solar-Terr. Phys., 61, 1123–1130.

Schmidt, T., Heise, S., Wickert, J., Beyerle, G., Reigber, C., 2005, GPS radio occul tation with CHAMP and SAC-C: Global monitoring of thermal tropopause parameters, Atmos. Chem. Phys, 5, 1473 ˇT1488

Son, S.-W., Tandon, N.F., Polvani, L.M, 2011, The fine-scale structure of the global tropopause derived from COSMIC GPS radio occultation measurements, J. Geophys. Res, 116, D20113, doi: 10.1029/2011JD016030

Tsuda, T., T. E. VanZandt, M. Mizumoto, S. Kato, and S. Fukao, Spectral analysis of temperature and Brunt Väisälä frequency fluctuations observed by radiosondes, J. Geophys. Res., 96, 17265–17278, 1991.

Wilcox L.J., Hoskins B.J., Shine K.P. 2012. A global blended tropopause based on ERA data. Part I: Climatology. Q. J. R. Meteorol. Soc. 138: 561–575. DOI:10.1002/qj.951.

Really thanks for recommending these valuable references.

Thank you again for your help with improving the paper.

Best regards

---

## Referee Comment (RC2) · Anonymous Referee #2 · 18 Mar 2019

In this study, the authors demonstrated the potential of MST radar in detecting the tropopause height (RT) from the radar backscattered echo power profiles by carrying out extensive comparison with the lapse rate tropopause (LRT) derived from radiosonde data and with dynamical tropopause (2 PVU) derived from ERA-Interim reanalysis dataset during the period Nov. 2011 to May 2017 covering all seasons. Comparison results showed good agreement between Radar and radiosonde and that between radar and ERA data in most of the seasons. The RT determination and comparison with other observations has been already carried out by many other investigators. However, a systematic comparison has been carried out in this paper and the difference in tropopause height is attributed to the sharpness of the tropopause inversion

layer (weak / strong). The potential of radar in examining the short-term variability of tropopause useful for wave studies etc and its limitation in detecting tropopause in few occasions are also discussed. In general, the paper is well written and the results are interesting. However, a few concerns need to be addressed, before the manuscript is published. 1) RT is determined using the vertical beam echo power data collected in "low mode" operation (which receives strong signal up to 14-15 km). In "middle mode", strong signals can be obtained in the altitude region 7-25 km (as seen in Fig. 8). Also, the "first tropopause" and "second tropopause" (based on WMO definition of LRT) are clearly evident in the mean effective data acquisition data obtained from middle mode operation (Fig.8). I strongly believe, that if "middle mode" vertical beam data is used, the strong gradients in radar echo power could be discernible corresponding to the altitudes of first and second tropopause. The authors can examine this aspect for available dataset in middle mode observations and compare with the first and second tropopause derived from radiosonde data. 2) Radar provides a vertical resolution of 150 m in "low mode" and 600 m in "middle mode" and "1200 m" in "high mode" and the temporal resolution is about 30 minutes. In the present study, RT derived from the vertical beam data in low mode is compared with the dynamical tropopause (2 PVU) derived from potential vorticity obtained from ERA-interim reanalysis The comparison results shows large deviation between the two. Fine resolution radar data is compared with the coarse resolution ERA dataset. What is sanctity in comparing these two datasets.

Specific/Minor comments

Line 28 : replace "good capability of Beijing MST radar" with "potential of Beijing MST radar "

Lines 108, 246, Fig. 8: Is this the "data acquisition rate" of backscattered echo power received ?. Effective data acquisition rate for different modes of radar operation are shown? How is this parameter estimated . Give details.

Lines 165-168: The method of identifying dynamical tropopause from potential vorticity is to be added. ERA-interim reanalysis data does not have fine vertical resolution. But the dynamical tropopause determined from the above is compared with RT derived from higher vertical resolution radar backscattered echo power. Hence, the larger difference in tropopause height is expected between the two methods.

Line 181: delete "fine-scale" Line 190-191: ".......the RT is well defined as the first layer with enhanced echo power..." Line 209: replace "good capability" with "potential"

Line 217: "sharpness of tropopause" is affected by cyclonic /anticyclonic systems. Explain. Are radar measurements carried out during such systems. please clarify.

Line 237, 246: what is "effective data acquisition rate?: Middle mode observation in Figure 8 shows two distinct peaks corresponding to the mean of first and second tropopauses based on LRT definition by WMO. Then why the data obtained from this mode (middle mode) is not used for the extensive comparison of first and second tropopause derived from radiosonde dataset, which is not so far studied extensively.

Lines 247-249: Correct this sentence (message not clear).

Line 272: "....radar-derived winds are combined...." what does it mean? Line 289-290: correct the sentence

Line 293-294: what are the system problems that makes RT identification difficult?

Lines: 297-298: correct the sentence Line 300: In this case, the temperature inversion is observed at 16 km...

Line 307-308: Correct this sentence....

Line 311: ..difficult in identifying the thermal tropopause from radiosonde profiles ..

Line 313: ...altitude extent of inversion layer is too thin to meet the WMO criterion...

Line 316: delete "Need to highlight again that"

Line 324: inconsistency between the RT and PVT

Line 326-327: Confirm whether radar measurements are carried out during cyclones /anticyclones in the upper troposphere (which period/season). Is the asymmetric differences in tropopause heights mainly due to the above conditions or due to difference in vertical resolution of radar and ERA dataset.

---

## Author Comment (AC4) · 18 Mar 2019

In this study, the authors demonstrated the potential of MST radar in detecting the tropopause height (RT) from the radar backscattered echo power profiles by carrying out extensive comparison with the lapse rate tropopause (LRT) derived from radiosonde data and with dynamical tropopause (2 PVU) derived from ERA-Interim re-analysis dataset during the period Nov. 2011 to May 2017 covering all seasons. Com-parison results showed good agreement between Radar and radiosonde and that be-tween radar and ERA data in most of the seasons. The RT determination and comparison with other observations has been already carried out by many other investigators. However, a systematic comparison has been carried out in this paper and the differ-ence in tropopause height is attributed to the sharpness of the tropopause inversion layer (weak / strong). The potential of radar in examining the short-term variability of tropopause useful for wave studies etc and its limitation in detecting tropopause in few occasions are also discussed. In general, the paper is well written and the results are interesting. However, a few concerns need to be addressed, before the manuscript is published.

**Response:**

Dear reviewer, we really thank you for the helpful and constructive comments, which will be of great useful for this article. We hope that the reviewers will be satisfied with our responses and revisions. Our responses are in different color style (reviewer's comments are shown in black and our response in blue type).

1) RT is determined using the vertical beam echo power data collected in "low mode" operation (which receives strong signal up to 14-15 km). In "middle mode", strong signals can be obtained in the altitude region 7-25 km (as seen in Fig.8). Also, the "first tropopause" and "second tropopause" (based on WMO definition of LRT) are clearly evident in the mean effective data acquisition data obtained from middle mode operation (Fig.8). I strongly believe, that if "middle mode" vertical beam data is used, the strong gradients in radar echo power could be discernible corresponding to the altitudes of first and second tropopause. The authors can examine this aspect for available dataset in middle mode observations and compare with the first and second tropopause derived from radiosonde data.

**Response:**

Yes, you are right. Really thanks for your valuable comments. It is necessary to explain here the concerns about the radar tropopause detection using middle mode data.

Firstly, the middle mode data is not appropriate to be used to detect the clear tropopause structure (both the first or the second tropopause). Figure R1 (shown below) shows the

middle mode observation results of the altitude-time intensity plot of radar backscattered echo power on February 2014. The month is the same as that in Figure 3 of the manuscript. Indeed, the first tropopause structure can be seen with middle mode observations, but the boundary is unsharpness and too coarse to identify the clear tropopause height, at least (especially) compared to the Figure 3 in the manuscript. In addition, also is the most important feature, the second tropopause is barely detected by middle mode results. The limited altitude resolution (600 m) and the limited radar transmitted power are likely the main causes.

[Figure]

Figure R1. Middle mode observation results: Altitude-time intensity plot of radar backscattered echo power for February 2014. '+' indicates the first tropopause; and 'x' denotes the higher second tropopause derived from radiosonde data.

Secondly, why the "first tropopause" and "second tropopause" appeared to be clear in the mean effective data acquisition rate (profile) obtained from middle mode operation (Fig.8)? We believe that this is largely because the data acquisition rate is the statistical result of five-year dataset, even if the height resolution is relatively low (600 m), the 5-year statistics are enough to amplify the impact of changes in atmospheric states (such as the static stability) around the transition region (between the troposphere and the stratosphere) on the data acquisition rate.

Finally, some responses regarding to the second tropopause:
Considering that the mechanism of the formation of second tropopause (Pan et al., 2004; Randel et al., 2007; Pan et al., 2009), the inversion height of ~16km (or higher) over the radar station is the second tropopause with tropical characteristics. Anyhow, no matter whether the inversion layer at ~16 km is the first tropopause or the second tropopause, such tropical featured higher tropopause will not be considered and studied here by both the RT definition and LRT definition. Indeed, the routine presented higher tropopause (second tropopause near 16km) in different seasons throughout the year is

worthwhile for studying. However, due the relatively poor altitude resolution for middle mode data, it is not appropriate to be used to detect high resolution tropopause structure over Beijing MST radar station, especially for the statistical study. The case observation of the second tropopause near 16 km (or higher) using the middle mode data is worthy of future study.

Therefore, we explain in many places that this study only focuses on the first tropopause below 16km, no matter whether it exists or not. For example, one sentence has been modified in the introduction section of the revised manuscript: 'In the present study, we focus only on the first tropopause (below 16 km) which will be referred to as 'tropopause' hereafter'.

Pan, L. L., Randel, W. J., Gary, B. L., Mahoney, M. J., and Hintsa, E. J.: Definitions and sharpness of the extratropical tropopause: A trace gas perspective. Journal of Geophysical Research, 109, D23103, doi:10.1029/2004JD004982, 2004.

Pan, L. L., W. J. Randel, J. C. Gille, W. D. Hall, B. Nardi, S. Massie, V. Yudin, R. Khosravi, P. Konopka, and D. Tarasick: Tropospheric intrusions associated with the secondary tropopause, Journal of Geophysical Research, 114, D10302, 2009.

Randel, W. J., Seidel, D. J., and Pan, L. L.: Observational characteristics of double tropopauses. Journal of Geophysical Research, 112, D07309, 2007.

2) Radar provides a vertical resolution of 150 m in "low mode" and 600 m in "middle mode" and "1200 m" in "high mode" and the temporal resolution is about 30 minutes. In the present study, RT derived from the vertical beam data in low mode is compared with the dynamical tropopause (2PVU) derived from potential vorticity obtained from ERA-interim reanalysis The comparison results shows large deviation between the two. Fine resolution radar data is compared with the coarse resolution ERA dataset. What is sanctity in comparing these two datasets.

**Response:**

Dear reviewer, really thanks for your comments. The difference in height resolution between radar and reanalysis data is unlikely to be one reason for the large difference between RT and PVT. There are also differences in resolution between radar and radiosonde data. At least, the difference in height resolution is not the main point. The interesting features from the comparison results between RT and PVT are that: the RTs agree reasonably well with the PVTs with the correlation coefficient of 0.72 and 0.76 respectively, during winter and spring (Fig. 6a and 6b). In contrast, the consistency for summer (Fig. 6c) is relatively bad and with correlation coefficient of only 0.33.

Whereas, in contrast, previous research about the RT and PVT results over polar regions by Alexander et al., (2013) reported that the comparison between the RT and PVT showed the similar good agreement during both summer and winter.

The possible causes of the larger offsets during summer (Fig 6c) is discussed in the revised manuscript. Following sentences in the revised manuscript (discussion section) have been added: 'The existing cyclones or anticyclones in the upper-troposphere (Wirth, 2000), of course, may also be an important cause of the significant asymmetric differences (most of the scattered points deviate significantly from the 1:1 line). This asymmetric differences, that is most of the RT are located higher than the 2PVU tropopause height, suggest that the 2PVU surface is not the best measure of a dynamical tropopause over Beijing during summer-time.'.

*Alexander, S.P., Murphy, D.J., and Klekociuk, A.R.: High resolution VHF radar measurements of tropopause structure and variability at Davis, Antarctica (69° S, 78° E), Atmos. Chem. Phys., 13, 3121-3132, 2013.*

Specific/Minor comments

Line 28 : replace "good capability of Beijing MST radar" with "potential of Beijing MST radar "
**Response:**
Really thanks for pointing out the flaw. The corresponding sentence has been replaced.

Lines 108, 246, Fig. 8: Is this the "data acquisition rate" of backscattered echo power received?. Effective data acquisition rate for different modes of radar operation are shown? How is this parameter estimated. Give details.
**Response:**
Really thanks for your comments. Data acquisition rate indicates the effective wind data. It has been corrected in the 3.3 section and the figure caption in the revised paper.

Lines 165-168: The method of identifying dynamical tropopause from potential vorticity is to be added. ERA-interim reanalysis data does not have fine vertical resolution. But the dynamical tropopause determined from the above is compared with RT derived from higher vertical resolution radar backscattered echo power. Hence, the larger difference in tropopause height is expected between the two methods.
**Response:**
Dear reviewer, the difference in height resolution between radar and reanalysis data is unlikely to be one key reason for the large difference between RT and PVT. There are also differences in resolution between radar and radiosonde data. At least, the difference in height resolution is not the main point. The possible causes of the larger offsets during summer (Fig 6c) is discussed in the revised manuscript:
'The existing cyclones or anticyclones in the upper-troposphere (Wirth, 2000), of course, may also be an important cause of the significant asymmetric differences (most of the scattered points deviate significantly from the 1:1 line). This asymmetric

differences, that is most of the RT are located higher than the 2PVU tropopause height, suggest that the 2PVU surface is not the best measure of a dynamical tropopause over Beijing during summer-time.'.

Line 181: delete "fine-scale"
**Response:**
Really thanks for pointing out the flaw. It has been corrected in the revised manuscript.

Line 190-191: ".......the RT is well defined as the first layer with enhanced echo power..."
**Response:**
It has been corrected in the revised manuscript.

Line 209: replace "good capability" with "potential"
**Response:**
It has been corrected in the revised manuscript.

Line 217: "sharpness of tropopause" is affected by cyclonic /anticyclonic systems. Explain. Are radar measurements carried out during such systems. please clarify.
**Response:**
Dear reviewer, we didn't demonstrat that the sharpness of tropopause is affected by cyclonic /anticyclonic systems. The results form Figure 5 indicate that the larger (weaker) tropopause sharpness contribute to lower (higher) difference between the RT and LRT.

Line 237, 246: what is "effective data acquisition rate?: Middle mode observation in Figure 8 shows two distinct peaks corresponding to the mean of first and second tropopauses based on LRT definition by WMO. Then why the data obtained from this mode (middle mode) is not used for the extensive comparison of first and second tropopause derived from radiosonde dataset, which is not so far studied extensively.
**Response:**
Really thanks for your comments. Data acquisition rate indicates the effective wind data. It has been corrected in the 3.3 section and the figure caption in the revised paper. Firstly, no matter whether the inversion layer at ~16 km is the first tropopause or the second tropopause, such tropical featured higher tropopause will not be considered and studied here by both the RT definition and LRT definition. Secondly, the middle mode data is not appropriate to be used to detect the clear tropopause structure (both the first or the second tropopause, please see Figure R1 above). Detailed responses about the issue of second tropopause are given above.

Lines 247-249: Correct this sentence (message not clear).
**Response:**
Really thanks for your comments. It has been corrected in the revised manuscript.

Line 272: "....radar-derived winds are combined...." what does it mean?

**Response:**

Really thanks for pointing out the flaw. The corresponding sentence has been corrected in the revised manuscript and modified to 'With the absence of temperature measurements, zonal and meridional winds are applied to demonstrate the evidence of diurnal or semidiurnal modulation by tidal'.

Line 289-290: correct the sentence

**Response:**

Really thanks for pointing out the flaw. The corresponding sentence has been corrected in the revised manuscript.

Line 293-294: what are the system problems that makes RT identification difficult?

**Response:**

The corresponding sentence has been cahgned to 'Apart from the system problems (e.g. the damage of T/R module)' in the revised manuscript.

Lines: 297-298: correct the sentence

**Response:**

It has been corrected in the revised manuscript.

Line 300: In this case, the temperature inversion is observed at 16 km...

**Response:**

Detailed responses about the issue of second tropopause are given above. In order to avoid the potential misguidance and to fit in with the main research focus of this paper, we have indicated in many places that the research focus of this paper is the first tropopause below 16km (as long as it exists).

For example, one sentence has been modified in the introduction section of the revised manuscript: 'In the present study, we focus only on the first tropopause (below 16 km) which will be referred to as 'tropopause' hereafter'.

In addition, the figure 12 and the figure caption have also been modified accordingly:

[Figure]

**Figure 12.** Example profiles of radar echo power and radiosonde temperature that (a) both the RT and LRT definitions fail due to the continuing decrease in temperature on 00 UTC 7 July 2012 and (b) the temperature inversion layer failed to meet the LRT definition but well defined in RT definition on 12 UTC 02 August 2012. Please note that we only consider the conditions below 16 km.

Line 307-308: Correct this sentence....
**Response:**
Really thanks for pointing out the flaw. The corresponding sentence has been corrected

Line 311: ..difficult in identifying the thermal tropopause from radiosonde profiles ..
**Response:**
Really thanks for your comments. It has been corrected in the revised manuscript.

Line 313: ...altitude extent of inversion layer is too thin to meet the WMO criterion...
**Response:**
Really thanks. It has been corrected in the revised manuscript.

Line 316: delete "Need to highlight again that"
**Response:**
Really thanks. It has been deleted in the revised manuscript.

Line 324: inconsistency between the RT and PVT
**Response:**
Really thanks. It has been corrected in the revised manuscript.

Line 326-327: Confirm whether radar measurements are carried out during cyclones/anticyclones in the upper troposphere (which period/season). Is the asymmetric differences in tropopause heights mainly due to the above conditions or due to difference in vertical resolution of radar and ERA dataset.

**Response:**

Dear reviewer, the difference in height resolution between radar and reanalysis data is unlikely to be one key reason for the large difference between RT and PVT. There are also differences in resolution between radar and radiosonde data. At least, the difference in height resolution is not the main point. Because the differences during spring and winter is not so bad as that during summer. The possible causes of the asymmetric differences during summer (Fig 6c) is discussed in the revised manuscript:
'The existing cyclones or anticyclones in the upper-troposphere (Wirth, 2000), of course, may also be an important cause of the significant asymmetric differences (most of the scattered points deviate significantly from the 1:1 line). This asymmetric differences, that is most of the RT are located higher than the 2PVU tropopause height, suggest that the 2PVU surface is not the best measure of a dynamical tropopause over Beijing during summer-time.'.

*Wirth, V.: Thermal versus dynamical tropopause in upper-tropospheric balanced flow anomalies. Quarterly Journal of the Royal Meteorological Society, 126(562), 299-317, 2000.*

Thank you again for your help with improving the paper.

Best regards

---

## Author Response (AR2)

**Topical Editor Decision: Publish subject to revisions (further review by editor and referees)**

(30 May 2019) by Andrew J. Kavanagh

Comments to the Author:

Reviewer 1 is broadly happy with your responses and why you do not identify the 2nd troposphere. However they insist that you complete the four key points that they have identified prior to publication.

Reviewer 2 is less satisfied with your responses. The main sticking point appears to be your lack of discussion of a 2nd troposphere using middle mode. I think it is essential that you include a clear discussion of this in the paper including why the measurement is not possible for the radar. Reviewer 1 has accepted this, but I think reviewer 2 is concerned that the general reading audience might not appreciate the problem.

Once you have made the required changes I will reassess the manuscript and determine whether the reviewers really need to see it again (I would hope not if you do everything they now ask).

**Dear Editor Kavanagh,**

We thank you and both reviewers for the constructive comments, which as outlined have helped improve the manuscript. I totally agree with your points. Especially, according to the comments of reviewer 1 and reviewer 2, it is really necessary to explain clearly in the revised manuscript that the Beijing MST radar cannot detect tropopauses with altitudes above 16km, whether or not these are the first or second tropopause.

In addition, in order to make the 'cyan asterisks' in Figure 3 visible, the asterisks have been plotted in white in revised manuscript. The modified Figure 3 is given below.

[Figure]

Detailed responses to the reviewers are given as follows. The revised manuscript with tracked changes is also attached later.

**Response to reviewer #1**

Second review of 'High-resolution Beijing MST radar detection of tropopause structure and variability over Xianghe (39.75° N, 116.96° E), China', by Chen et al.

Note that in my comments below, I refer to the manuscript which the authors have incorporate both Reviewers' comments from the first review round.

I thank the authors for considering and addressing my comments. It is now clear why Chen et al. have had to reject consideration of tropopauses with altitudes above 16km, whether or not these are the first or second tropopause. This is a pity, but clearly a limitation of the radar system's power and capabilities.

As such, once the authors have implemented the following minor points, I would find this manuscript acceptable for publication.

**Dear Reviewer,**

We really thank the reviewer for the second encouraging view of our work.

1) In your abstract, you need to clearly state that you only consider the lower tropopause altitudes. This is not mentioned at all at present but is a very important point to include in the abstract. I suggest words on line 19 such as: "We only consider tropopause altitudes below 16km in this study because of limitations with the radar system", or words to this effect.

**Response:**

Complied with, we agree that this is necessary to be mentioned in the abstract. Words " We only consider tropopauses below 16km in this study because of limitations with the radar system" has added in the abstract.

2) Line 214 describing Figure 5, I think a better way to say this is: 'Higher probabilities of large tropopause sharpness values occur when the RT-LRT difference is less than 0.5km"

**Response:**

Complied with the suggestion. The corresponding sentence describing Figure 5 has been changed to 'What is apparent is that higher probabilities of large tropopause sharpness values occur when the RT-LRT difference is less than 0.5km'.

3) Line 248: I can't see from Figure 7 that the 'radar power points' show less dispers[ion]. Please remove the words 'and radar power points'

**Response:**

Thanks for pointing this out. Complied with your suggestion and we have removed the words 'and radar power points' in the revised manuscript.

4) Figure 3. The 'cyan asterisks' which indicate the LRT location are practically invisible because you are plotting a light blue color on to a dark blue color. I suggest you plot these asterisks in white, gold, or bright red. That should make them visible.

**Response:**

Thanks for your suggestion. The asterisks have been plotted in white in the revised manuscript.

**Response to reviewer #2**

I appreciate the effort of authors in revising the manuscript and providing good responses.

However, I am not totally satisfied with the revised version. Main concern is on the identification of second tropopause. This should be addressed by middle mode radar observations. Reviewer#1 also suggested to examine both the thermal tropopause (first and second LRT) using low mode and middle mode MST radar observations.

Though the focus of the study is on first troposphere, the authors should attempt to analyze the middle mode radar observations to identify the second tropopause, if exist. This should be added and discussed in the manuscript. Adding this would be a significant contribution to science/literature.

**Dear Reviewer,**

We really thank you for your helpful and constructive comments. If the second tropopause is present and sharpness enough, it should indeed be tried to detect the second tropopause using middle mode data. However, because of the limitations of the radar system itself (transmitted power, height resolution, etc.), second tropopauses with altitudes near 16 km or higher are not considered in this study.

In our first responses, we have given one month observation result (the month is same as that of Figure 3), which repeated here.

[Figure]

Figure R1. Middle mode observation results: Altitude-time intensity plot of radar backscattered echo power for February 2014. '+' indicates the first tropopause; and 'x' denotes the higher second tropopause derived from radiosonde data.

The results show that the first tropopause structure can be seen with middle mode observations, but the boundary is unsharpness and too coarse to identify the clear tropopause height, at least (especially) compared to the Figure 3 in the manuscript. In addition, also is the most important feature, the second tropopause is barely detected by middle mode results.

In order to explain clearly in the revised manuscript that the Beijing MST radar cannot detect tropopauses (whether or not these are the first or second tropopause.) with altitudes above 16km, words "
[revised manuscript text omitted]

---

## Author Response (AR3)

**Dear Editor Kavanagh,**

We thank you and both reviewers for the constructive comments, which as outlined have helped improve the manuscript.

My second affiliation have been added in the complete manuscript, because I am currently a teacher in Nanchang Hangkong University.

Thank you very much again.